

# In Situ Microphysics Observations of Intense Pyroconvection from a Large Wildfire

David E. Kingsmill[1], Jeffrey R. French[2], and Neil. P. Lareau[3]

[1]Cooperative Institute for Research in Environmental Studies, University of Colorado, Boulder, Colorado, USA
[2]Department of Atmospheric Science, University of Wyoming, Laramie, Wyoming, USA
[3]Department of Physics, University of Nevada, Reno, Reno, Nevada, USA

*Correspondence to*: David Kingsmill (David.Kingsmill@colorado.edu)

**Abstract.**

This study characterizes the size and shape distributions of 10 μm to 6 mm diameter particles observed during six
penetrations of wildfire-induced pyroconvection near Boise, Idaho, USA by a research aircraft over the period 29-30 August
2016. In situ measurements by the aircraft include winds, atmospheric state, bulk water content and particle concentration,
size, and shape. These observations are complemented by data from airborne and ground-based radars. One of the
penetrations is through a subsaturated smoke/ash plume with negligible cloud liquid water content that is characterized by an
updraft of almost 36 m s$^{-1}$. The size distribution of number concentration is very similar to that documented previously for a
smoke plume from a prescribed fire and particle shapes exhibit qualitative and quantitative attributes comparable to ash
particles created in a burn chamber. Particles sampled during this penetration are most likely pyrometeors composed of ash.
Pyrocumulus clouds are probed in the other penetrations where values of relative humidity and cloud liquid water content are
larger, but updrafts are weaker. Compared to the smoke-plume penetration, size distributions are mostly characterized by
larger concentration and particle shapes exhibit a higher degree of circular symmetry. Particle composition in these
pyrocumulus penetrations is most likely a combination of hydrometeors (ice particles) and pyrometeors (ash).

## 1 Introduction

Wildfires are one of the most impactful natural hazards across the globe. Along with the injuries, loss of life and destruction
of property that wildfires directly influence, emissions from wildfires can create significant negative health effects, even at
distances far from and times long after the source emissions (e.g., Johnston et al., 2012; Thelen et al., 2013; Xu et al., 2020;
O'Dell et al., 2020). Not surprisingly, the corresponding economic impacts of wildfires are substantial (e.g., Ashe et al.,
2009; Jones et al., 2016; Jones and Berrens, 2017; Wang et al., 2020). These impacts have grown over the last several
decades due to a trend of increasing wildfire frequency, size, and severity (e.g., Westerling et al., 2006; Dennison et al.,
2014; Westerling, 2016; Parks and Abatzoglou, 2020). This trend is often attributed to enhanced fuel aridity brought about
by anthropogenic global warming (e.g., Flannigan et al., 2009; Yue et al., 2013; Barbero et al., 2015; Abatzoglou and
Williams, 2016).



Pyroconvection occurs when fire-released heat, moisture and/or aerosols induce or augment convection in the atmosphere (McCarthy et al., 2018). One manifestation of pyroconvection is the plume of smoke and ash that a wildfire generates. If the heat produced by a wildfire is sufficiently intense and atmospheric conditions are appropriate, smoke/ash plumes can rise to a level where they become saturated with water vapor and form pyrocumulus clouds (American

Meteorological Society, 2022a). Pyrocumulus clouds (pyroCus) are sometimes capable of developing into pyrocumulonimbus clouds (pyroCbs) that can generate precipitation, downdrafts, and lightning (American Meteorological Society, 2022b). Some particularly intense pyroCbs have been associated with fire-induced tornadoes (Fromm et al., 2006; Lareau et al., 2018, 2022).

Wildfire-induced pyroconvection contains particles with a diversity of composition, phase (in the case of

hydrometeors), size, and shape. The most comprehensive documentation of these particles has been for smoke particulates less than ~20 µm diameter within plumes at various distances downwind of wildfires (e.g., Radke et al., 1978; Radke et al., 1991; Reid et al., 2005; Kleinman et al., 2020). A large fraction (80-90%) of the smoke aerosol particles are in the accumulation mode (0.1-2 µm diameter) while a smaller fraction (~10%) are in the coarse mode (2-20 µm diameter). These smoke particulates are primarily composed of carbonaceous material (50-60% organic carbon and 5-10% black carbon) and

exhibit a variety of forms such as chain aggregates, solid irregulars, and spherical shapes. Most observations of these aerosol particles have been achieved with in situ sampling by research aircraft. However, four relatively recent studies employed satellite remote sensing to document smoke particulates (Jethva and Torres, 2011; Konovalov et al., 2017; Junghenn-Noyes et al., 2020a,b). These relatively small-sized smoke particulates are responsible for most of the negative health effects from wildfire emissions.

The size of cloud droplets in pyroCu and pyroCb has also received some attention. Andreae et al. (2004) report in situ observations of droplets in pyroconvective clouds characterized by a modal diameter of 12 µm with a distribution tail extending to ~40 µm. In addition, Rosenfeld et al. (2007) use satellite observations of a violent pyroCb to retrieve cloud-droplet effective radii whose median values did not exceed 11 µm. Results from both studies are consistent with the notion that smoke from biomass burning can lead to smaller cloud droplet sizes, which has been hypothesized to inhibit

precipitation (Rosenfeld and Lensky, 1998; Rosenfeld, 1999).

Another category of high-impact particles contained within pyroconvection is firebrands, which are actively burning pieces of combustion debris that can be transported up to several tens of kilometres downstream of the main fire front to initiate spot fires and facilitate rapid fire spread (Williams, 1982; Koo et al., 2010). Firebrands are significantly larger than smoke particulates in plumes and cloud droplets in pyroCu and pyroCb. Laboratory studies have quantified firebrand size

and shape characteristics in association with the burning of individual Douglass fir and Korean pine trees (Manzello et al., 2007 and Manzello et al., 2009, respectively). Cylindrically shaped combustion debris of 30-50 mm length and 3-5 mm diameter are produced. Field studies have also been employed to document firebrand characteristics in the aftermath of experimental fires (El Houssami et al., 2016; Thomas et al., 2017; Filkov et al., 2017) and wildfires (Manzello and Foote,



2014) that consumed coniferous forest. Results from these efforts indicate the presence of cylindrically shaped combustion
debris more than 30 mm length, but they occur in relatively low numbers. Most firebrands have a less elongated, irregular
shape with maximum dimensions of 5-30 mm. Remote sensing has not yet been employed to characterize the size and shape
of firebrands. However, results from these in situ studies have informed the use of ground-based radar to detect and track
firebrands (McCarthy et al., 2019b).

The characteristics of pyroconvection particles smaller than firebrands (i.e., diameter $\lesssim$ 5 mm) but larger than smoke
particulates and cloud droplets (i.e., diameter $\gtrsim$ 20 μm) has been explored to a much lesser extent. In their review of
biomass burning emissions, Reid et al. (2005) mentions the existence of giant ash particles with diameters greater than
20 μm, extending up to and exceeding 1 mm. Likewise, Andreae et al. (2004) allude to giant ash particles with diameters
that range from sub millimeter to a few centimeters. However, neither study provides direct evidence to support these
assertions. To our knowledge, the only explicit, in situ documentation of ash particles with diameters larger than ~20 μm is
contained in Fig. 28.6 of Radke et al. (1991). This figure shows the number-concentration size distribution of ash particles
in association with a smoke/ash plume from a prescribed fire. The largest ash particles have diameters on the order of
several millimetres. Radke et al. (1991) state that the presence of such large ash particles was typical for all the large fires
they studied. Unfortunately, they provide only minimal additional information to contextualize these giant-ash observations.
For example, the location of the observations relative to the fire is not indicated and the dynamic character of the plume (i.e.,
depth, vertical motions) is not described. Additionally, the shapes of these particles are not documented.

With the dearth of in situ observations, other approaches have been employed to document giant ash particles. Baum et
al. (2015) use a laboratory burn chamber to create samples of ash from the combustion of messmate eucalypt biomass.
Shapes of the resulting ash particles are primarily planar or cylindrical and are characterized by areas ranging from 0.2 mm²
(the minimum detectable area for their imaging instrument) to ~20 mm². This range of particle area corresponds to an
equivalent circle diameter range of ~0.5 mm to ~5 mm, which is consistent with the diameters of giant ash particles observed
in smoke/ash plumes by Radke et al. (1991). The shapes of giant ash particles from biomass burning have also been
examined remotely using polarimetric radar observations of smoke/ash plumes (Banta et al., 1992; Melnikov et al., 2008;
Lang et al., 2014; Lareau and Clements, 2016; McCarthy et al., 2018; McCarthy et al., 2019b; Zrnic et al., 2020). These
observations, particularly differential reflectivity ($Z_{DR}$: difference between logarithmic reflectivity from horizontal and
vertical polarizations) and correlation coefficient ($\rho_{hv}$: correlation between horizontally and vertically polarized radar return
signals) suggest that giant ash particles fall with a horizontal orientation ($Z_{DR} \gtrsim$ 2 dB) and have irregular shapes ($\rho_{hv} \lesssim$ 0.4)

In situ measurements of pyroCu and pyroCb particles larger than a few tens of microns (i.e., cloud droplets) have only
recently been reported in the peer-reviewed literature. Peterson et al. (2022) document the size distributions of particles
from a few microns to a few millimetres in the middle to upper portions of active pyroCbs and adjacent detached anvils.
This sampling occurs at temperatures of -40°C, -31°C and -18°C. Particles larger than ~15 μm are assumed to be composed
of ice based on the qualitative character of a few optical array probe images. However, no quantitative analysis of particle



shape is provided. Analysis of pyroCu and pyroCb particles has more commonly been accomplished remotely with the use of polarimetric radar observations to discriminate between giant ash particles and rimed ice particles, such as graupel and hail (Lang et al., 2014; Lareau and Clements, 2016, McCarthy et al., 2018; McCarthy et al., 2019b). Specifically, graupel

and hail are thought to be characterized by relatively small $Z_{DR}$ and relatively large $\rho_{hv}$ compared to giant ash particles.

McCarthy et al. (2019a) introduced the term "pyrometeor" to describe all debris of a pyrogenic origin with diameter greater than or equal to a millimetre. The stated intent of this definition was to differentiate smoke particulates from larger pyroconvection particles capable of scattering transmitted radiation from radars operating at S-band (~10 cm wavelength), C-band (~5 cm wavelength) and X-band (~3 cm wavelength). Unfortunately, this definition excludes ash particles between

~20 μm diameter (i.e., the upper boundary of smoke particulates) and 1 mm. Therefore, this study defines the term pyrometeor as all debris of a pyrogenic origin with diameter greater than or equal to 20 μm. Pyrometeors include both ash particles and firebrands and are distinct from hydrometeors (liquid or solid water particles).

As the preceding discussion has highlighted, in situ observations of pyroconvection particles are lacking, especially for particles larger than smoke particulates and cloud droplets. McCarthy et al. (2019a) emphasized this point in their review of

the use of weather radar for wildfire research and described one reason why it is so problematic. Specifically, they noted that the scarcity of these observations has impeded the application of radar to study pyroconvection caused by wildfires. This dearth of in situ measurements also negatively impacts the validation and improvement of various models employed to simulate wildfires and related pyroconvection (e.g., Coen et al., 2013; Peace et al., 2015; Kochanski et al., 2016; Toivanen et al., 2019). The present study takes steps toward addressing these issues through analysis of in situ microphysics data

collected by a research aircraft during penetrations of pyroconvection over the period 29-30 August 2016 (Clements et al., 2018; Rodriguez et al., 2020). This pyroconvection was caused by the Pioneer Fire, a large wildfire in the intermountain USA region northeast of Boise, Idaho. Size distributions of number concentration and area concentration are characterized for particles spanning the diameter range of 10 μm to 6 mm in smoke/ash plumes and pyroCu. Also, particle shapes are examined with both qualitative and quantitative approaches. Finally, all of these in situ observations are placed in context by

employing airborne Doppler radar observations to characterize the depth and vertical motions of the sampled pyroconvection.

## 2 Data and analysis methods

The principle observing platform used in this study is the University of Wyoming King Air (UWKA) research aircraft. A diverse collection of in situ and remote-sensing instrumentation was installed on the UWKA (Wang et al., 2012). Standard

flight level parameters are derived with in situ sensors that measure navigation (e.g., 3D position, ground speed, airspeed, orientation), winds (e.g., horizontal wind speed and direction, vertical air velocity) and atmospheric state (e.g., pressure, temperature, water vapor content). This study uses versions of these parameters temporally degraded to 1 Hz from higher rate raw data. In situ microphysics data used in this study can be divided into two different measurement types: bulk water





content and particle concentration, size, and shape (Table 1). Four different sensors are employed to characterize bulk water
content. Liquid water content (LWC) is quantified with the LWC-100 probe and the PVM-100 probe. The Nevzorov probe
is composed of two sensors: one to measure LWC and the other to measure total condensed water content (TWC). In
combination, the sensors on the Nevzorov probe quantify both LWC and ice water content (IWC). IWC can be subject to
much greater uncertainty than LWC or TWC since it is the difference between those measures. The measurement range,
manufacturer and relevant reference for these probes is listed in Table 1. Particle concentration, size, and shape are
characterized with three different optical array probes (OAPs). The Two-Dimensional Stereo (2D-S) probe provides particle
shadowgraph images in two orthogonal planes: one oriented vertically (2D-SV) and the other horizontally (2D-SH). Images
from the 2D-S cover the size range of 10-1280 μm, with 10 μm resolution. The Cloud Imaging Probe (CIP) produces
images at 25 μm resolution from 25-1600 μm while the Two-Dimensional Precipitation (2D-P) probe produces images at
200 μm resolution from 200-6400 μm. Both the CIP and 2D-P are oriented vertically.

| Measurement Type | Instrument | Measurement Range | Manufacturer | Reference |
|---|---|---|---|---|
| Bulk Water Content | LWC-100 | 0.05-3 g m$^{-3}$ | Droplet Measurement Technologies (DMT) | King et al. (1978) |
| | PVM-100 | 0.002-3 g m$^{-3}$ | Gerber Scientific Inc. (GSI) | Gerber et al. (1994) |
| | Nevzorov-LWC | 0.01-3 g m$^{-3}$ | Sky Phys. Tech. Inc. | Korolev et al. (1998) |
| | Nevzorov-TWC | 0.01-3 g m$^{-3}$ | Sky Phys. Tech. Inc. | Korolev et al. (1998) |
| Particle Concentration, Size and Shape | 2D-S | 10-1280 μm (10 μm) | Stratton Park Engineering Company (SPEC) | Lawson et al. (2006) |
| | CIP | 25-1600 μm (25 μm) | DMT | Baumgardner et al. (2001) |
| | 2D-P | 200-6400 μm (200 μm) | Particle Measuring Systems (PMS) | Knollenberg (1970) |

Table 1. Characteristics of UWKA in situ microphysics data used in this study. The measurement range for each instrument is listed. Size resolutions for the 2D-S, CIP and 2D-P instruments are indicated in parentheses.

Data from all three OAPs are quality-controlled and quantitatively processed with the University of Illinois/Oklahoma
OAP Processing Software (McFarquhar et al., 2018). First, artificial images are identified and rejected. These include zero
area images, split images, "streakers" (i.e., unrealistically long, and narrow images) and images resulting from ice particle
shattering (Field et al., 2006). The approaches to identify artificial images are based on experiences with sampling
hydrometeors. It is unclear whether sampling pyrometeors leads to the same types of artifacts. Accepted images are then
analyzed to determine their, diameter ($D$; diameter of minimum enclosing circle), aspect ratio ($ASP_r$; semi-minor axis length





divided by semi-major axis length), area ($A$), area ratio ($A_r$; area divided by area of minimum enclosing circle), and fine-detail ratio ($F_r$; Holroyd, 1987) defined as the perimeter times diameter divided by area. Distributions of number

concentration ($\widehat{N}$) and area concentration ($\widehat{A}$) as a function of diameter are derived for the size bins of each probe at 1-s resolution. Calculation of sample volume employs the reconstruction technique described by Heymsfield and Parrish (1978). The size distributions of $\widehat{N}$ are used to derive total concentration ($N_T$) and area-weighted mean diameter ($\overline{D}_{area}$):

$$N_T = \Delta D \sum_{i=1}^{i=max} \widehat{N}_i \qquad (1)$$

$$\overline{D}_{area} = \sum_{i=1}^{i=max} \widehat{N}_i D_i^3 \Big/ \sum_{i=1}^{i=max} \widehat{N}_i D_i^2 \qquad (2)$$

where $i$ is the size bin index and $\Delta D$ is the bin width, which is set to the resolution of each probe.

The UWKA remote sensing instrument used in this study is the Wyoming Cloud Radar (WCR), a W-band (~3.2 mm wavelength) pulsed Doppler radar. Data from two WCR antennae are employed: one pointed downward near nadir with a beamwidth of 0.5° and the other pointed upward near zenith with a beamwidth of 0.7°. Along track sampling of the data is at intervals of ~5 m and range resolution is ~15 m. Reflectivity from the WCR is sensitive to ~-35 dBZ$_e$ at 1 km range and

accurate to ~3 dBZ$_e$. The Doppler radial velocity is unambiguous over the range of ±15.8 m s$^{-1}$ (i.e., the Nyquist velocity). WCR data are edited to remove noise and ground clutter and de-aliased to correct folded radial velocities.

This study also employs data from a ground-based remote sensing instrument, namely the National Weather Service (NWS) Doppler radar located in Boise, Idaho (KCBX). Scans of reflectivity, Z$_{DR}$ and ρ$_{hv}$ at various elevation angles are used to provide context for the pyroconvection associated with the Pioneer Fire. Also, a volumetric "echo top" product is

utilized to characterize the maximum height of reflectivity more than 18 dBZ$_e$ for each column of range gates. Use of the 18 dBZ$_e$ threshold may underestimate the true tops of smoke/ash plumes since radar detections for ash are characteristically lower than for hydrometeors.

## 3 The Pioneer Fire and associated pyroconvection

The Pioneer Fire started on 18 July 2016 in the Boise National Forest (BNF) northeast of Boise and consumed a total of

76,081 ha (188,000 acres, 761 km$^2$) over the next 2 months. Terrain in the BNF varies from ~1 to ~2.5 km MSL (Fig. 1a) and vegetation is primarily composed of coniferous trees including ponderosa pine, Douglas fir and Engelmann spruce. This study focuses on the 29-30 August period when the Pioneer Fire rapidly advanced toward the northeast. Fire perimeters produced by the National Infrared Operations (NIROPS) Unit of the U.S. Forest Service and obtained from the National Interagency Fire Center (National Interagency Fire Center, 2016) show that the fire consumed 11,736 ha (29,000 acres, 117

km$^2$) in ~24 h.

The UWKA sampled pyroconvection associated with the Pioneer Fire from ~2230 UTC 29 August to ~0130 UTC 30 August (Fig. 1b), which represents late afternoon into early evening relative to local time. Maximum KCBX echo top heights during this period were ~12 km MSL and displaced northeast of the fire due to advection by the prevailing southwest winds (not shown). A photograph taken from the UWKA at 2238 UTC looking toward the northeast provides a visual

perspective of the pyroconvection (Fig. 1c). Smoke and ash-filled plumes emanating from the fire are evident at low levels. At higher levels, where some of the plumes become water saturated, pyroCu are apparent.

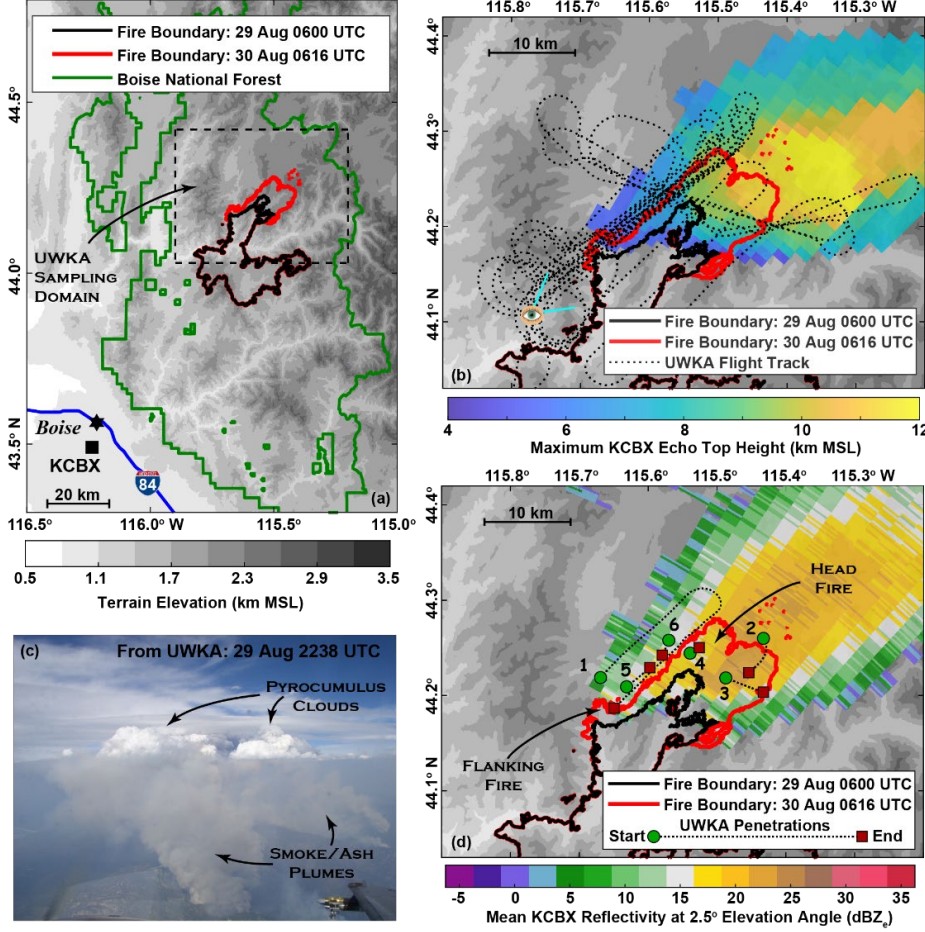

Figure 1. (a) Topographic map of the study area northeast of Boise, Idaho. Gray scale at bottom indicates terrain height. The boundary for the Boise National Forest is shown by green lines. Perimeters of the Pioneer Fire at 0600 UTC 29 August 2016 and 0616 UTC 30
August 2016 are indicated with black and red lines, respectively. Location of the KCBX radar is shown by the black-filled square. Black-dashed rectangle indicates the UWKA sampling domain shown in (b) and (d). (b) Maximum KCBX echo top heights for the period 2230 UTC 29 August 2016 to 0130 UTC 30 August 2016. Colour scale at bottom indicates plotted values. Terrain heights and fire perimeters from (a) are also shown. Black-dashed lines indicate the UWKA flight track for this period. (c) Photograph taken from the UWKA at 2238 UTC 29 August 2016. Location of the photograph is indicated by the eyeball in (b) and is directed towards the northeast. (d) Mean
KCBX reflectivity at 2.5° elevation for the period 2240 UTC 29 August 2016 to 0030 UTC 30 August 2016. Colour scale at bottom indicates plotted values. Terrain heights and fire perimeters from (a) are also shown. Locations of the head fire and flanking fire are indicated. Black-dashed lines show the locations of pyroconvection penetrations by the UWKA during this period, with green circles and red squares indicating the start and end of each penetration, respectively. Numbers next to each starting point indicate the numeric order of the penetrations (Table 2).





Six penetrations of pyroconvection were executed by the UWKA (Table 2). The first penetration went through a smoke/ash plume at 5.2 km MSL while the remaining five penetrations went through pyroCus at 7.3-7.7 km MSL. Locations of the penetrations relative to the fire provide additional context (Fig. 1d). The first three penetrations sample pyroconvection generated by the rapidly northeastward advancing area of the fire called the "head fire". Pyroconvection produced by a "flanking fire" that developed to the west-southwest of the head fire after penetration 1 is the focus of penetrations 4-6. Mean KCBX reflectivity at 2.5° elevation during the period of these penetrations shows maximum values of 20-25 $dBZ_e$ located southwest of the peak echo top heights. This displacement is reasonable given that the KCBX 2.5° beam height is ~6 km MSL at the position of the maximum values (a little lower to the southwest and a little higher to the northeast) and the maximum echo top heights are ~12 km MSL.

| Penetration | Start-End (UTC) | Mean Altitude (km MSL) | Pyroconvection Type | Location |
|---|---|---|---|---|
| 1 | 29 Aug 224040-224630 | 5.2 | Smoke/Ash Plume | Head Fire |
| 2 | 29 Aug 231230-231320 | 7.7 | Pyrocumulus | Head Fire |
| 3 | 29 Aug 233440-233520 | 7.3 | Pyrocumulus | Head Fire |
| 4 | 30 Aug 001750-001800 | 7.7 | Pyrocumulus | Flanking Fire |
| 5 | 30 Aug 002630-002700 | 7.7 | Pyrocumulus | Flanking Fire |
| 6 | 30 Aug 002840-002900 | 7.7 | Pyrocumulus | Flanking Fire |

Table 2. Time periods and altitudes of the six pyroconvection penetrations examined in this study. Pyroconvection type and location of the penetrations relative to the fire are indicated (see Fig. 1d)

**3.1 Penetration 1: Smoke/ash plume**

Penetration 1 involves a straight track toward the northeast followed by a 180° right turn that leads into a straight track toward the southwest (Fig. 1d). The turn of this penetration is at a range of ~105 km from KCBX, where the 1.5° elevation scan is at ~4.4 km MSL and the 2.5° elevation scan is at ~6.2 km MSL. These heights straddle the 5.2 km MSL UWKA flight altitude of the penetration. Reflectivity is significantly larger at 2.5° (Fig. 2a) compared to 1.5° (Fig. 2d), especially in the area near the turn where values reach 25 $dBZ_e$. The polarimetric fields ($Z_{DR}$ and $\rho_{hv}$) are relatively noisy, with large variations over short horizontal distances (Fig. 2b,c,e,f). $Z_{DR}$ in this area is as small as ~0 dB and a large as ~7 dB (Fig. 2b,e), which makes it difficult to conclude anything about particle orientation. $\rho_{hv}$ shows somewhat less variation, with most values less than 0.5 (Fig. 2c,f), which is suggestive of irregularly shaped particles.

Data from the WCR provides context for the vertical structure of pyroconvection sampled during this period (Fig. 3). The WCR is disabled until 224108 UTC and only the down antenna is in operation for the remainder of the plotted interval. Additionally, WCR data are not shown during the turn since pointing angles of the down antenna are far from nadir and vary significantly, making the data difficult to interpret.

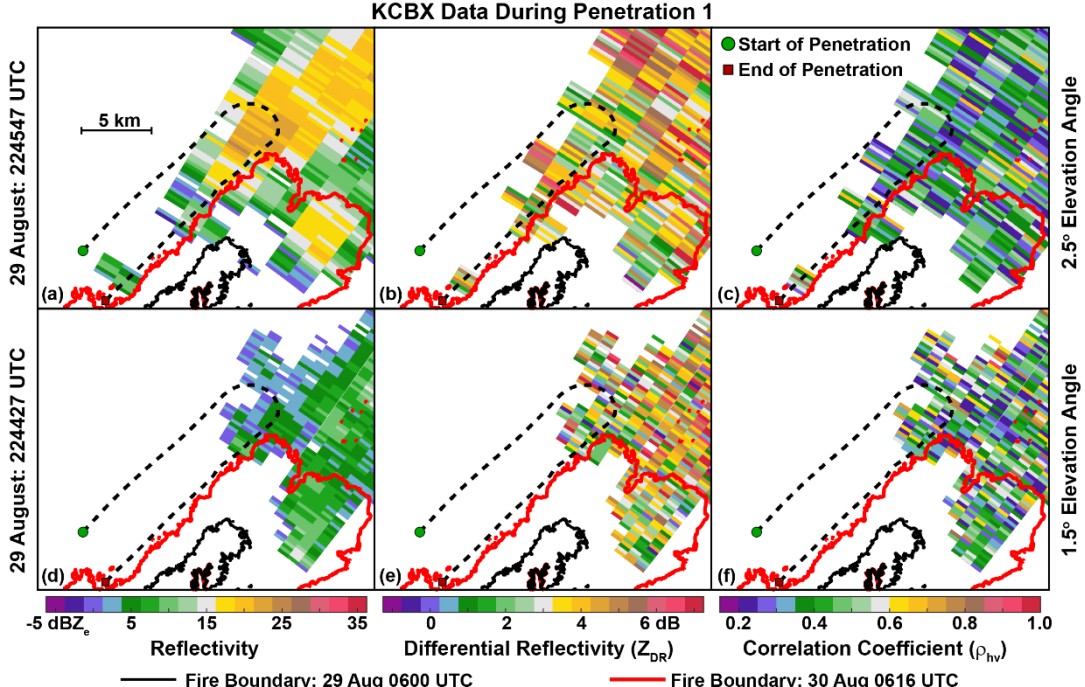

Figure 2. KCBX data approximately coincident with the execution of penetration 1. Left column (a,d) shows reflectivity, middle column (b,e) shows differential reflectivity ($Z_{DR}$) and right column (c,f) shows correlation coefficient ($\rho_{hv}$). Colour scales at bottom of each column indicate plotted values. Top row (a-c) is for data at 224547 UTC 29 August 2016 (2.5° elevation angle) and bottom row (d-f) is for data at 224427 UTC 29 August 2016 (1.5° elevation angle). Perimeters of the Pioneer Fire at 0600 UTC 29 August 2016 and 0616 UTC 30 August 2016 are indicated with black and red lines, respectively. Black-dashed lines show the location of penetration 1, with green circles and red squares indicating the start and end of the penetration, respectively.

WCR reflectivity is primarily less than -15 dBZ$_e$ prior to the turn but maximum values of -5–0 dBZ$_e$ are evident after the turn (Fig. 3a). The higher magnitude reflectivity cells are up to ~1 km wide and extend 0.5-2.5 km in depth. Some cells are tied to the surface (e.g., 224440 UTC and 224710 UTC) while others appear cut off from the surface and extend above 4 km MSL, often up to flight level. It is notable that maximum values of WCR reflectivity are -5–0 dBZ$_e$ while maximum values of KCBX reflectivity along penetration 1 after the turn are 15-20 dBZ$_e$ (Fig. 1d). Some of this discrepancy may be related to mismatched sample volumes, but that does not explain a 20 dBZ$_e$ offset. A more important factor may be that KCBX operates at a wavelength of ~10.7 cm (S-band) while the WCR operates at a wavelength of ~3.2 mm (W-band). Scattering characteristics of pyroconvection particles are likely different at these two wavelengths. Results from the present study will be used to address this issue in a future investigation.

WCR radial velocity from the down antenna is effectively vertical air velocity (Fig. 3b), with positive (negative) radial velocities toward (away from) the radar at flight level representing updrafts (downdrafts). This assumes that the fall speed of pyroconvection particles is negligible compared to the magnitude of vertical air motions. The magnitude of vertical velocity is significantly larger after the turn, especially for updrafts where values greater than 10 m s$^{-1}$ are common. Indeed, updrafts exceeding 20 m s$^{-1}$ are evident near flight level during the first 1-2 min after the turn. Most of the updraft maxima after the





turn are collocated with cells of higher magnitude reflectivity (Fig. 3a). Downdrafts are apparent, but their magnitude is smaller than that for updrafts (Fig. 3b). Also, downdrafts are more limited in vertical extent compared to updrafts.

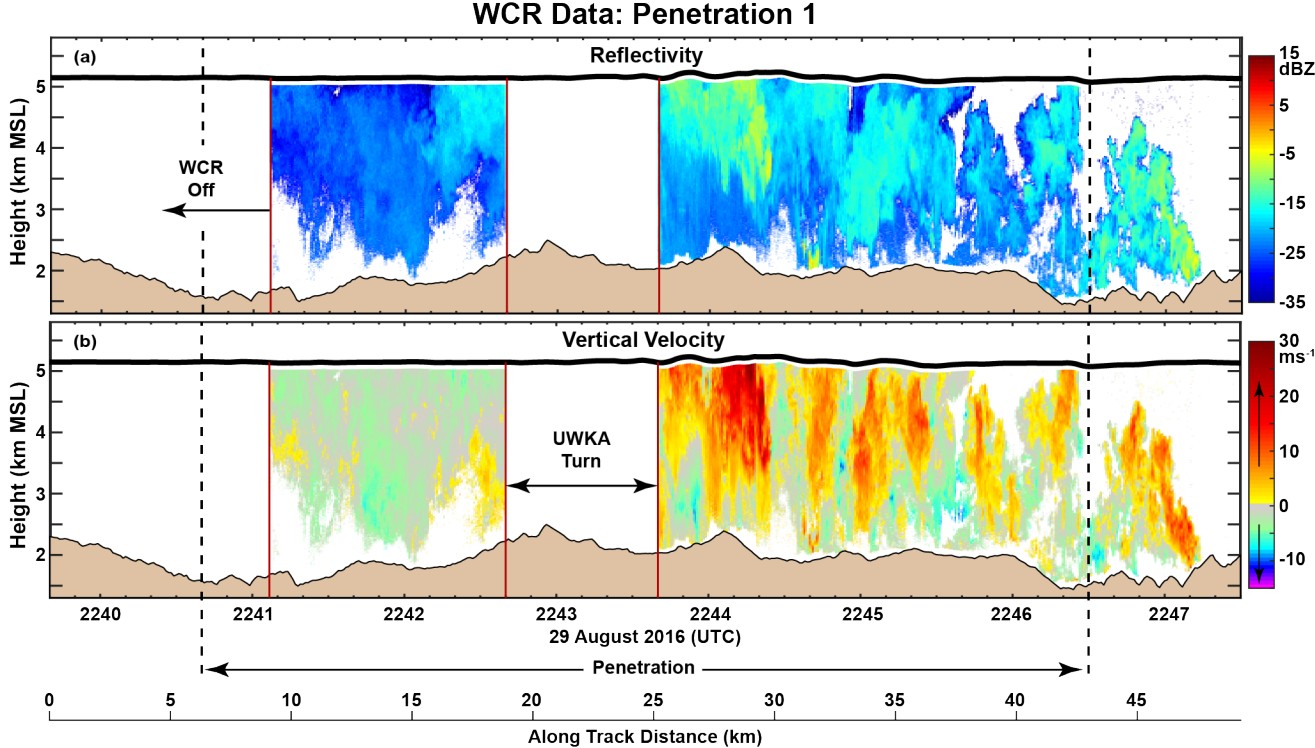

Figure 3. UWKA WCR (a) reflectivity and (b) radial velocity (effectively vertical velocity) data for penetration 1 (Fig. 1d, Table 2) and 60 s before and after the penetration (bounded by vertical, black-dashed lines). Colour scales to the right of each panel indicate plotted
values. Aircraft track shown by the bold black lines near the top of each panel. Underlying topography indicated by the tan-filled outline at the bottom of each panel. Along-track distance for the penetration shown at bottom. Periods without WCR data bounded by vertical red lines.

Vertical air velocity at flight level (Fig. 4a) is negative during most of the period before the turn, with downdrafts as large as 5 m s$^{-1}$. Thereafter on the penetration, vertical motions are mostly upward and more erratic, especially 224300-
224500 UTC. This period is highlighted by two updraft peaks of more than 16 m s$^{-1}$ (224340-224400 UTC) and a single updraft peak of almost 36 m s$^{-1}$ (224400-224420 UTC). These peaks in flight-level vertical air velocity occur at the same times that strong updrafts are evident near flight level in the WCR radial velocity data (Fig. 3b). Air temperature during the penetration varies between ~-6° C and ~-2°C (Fig. 4b). The warmest air is coincident with the updraft peaks just after the turn and suggests that the updrafts are associated with positive buoyancy (Rodriguez et al. 2020). Relative humidity is less
than 80% across the entire penetration and decreases to as little as 20% at some points. Also, bulk water content (Fig. 4c) is very small, with values that are near or below the sensitivity thresholds for the Nevzorov, LWC-100 and PVM-100 probes (Table 1). The subsaturated relative humidity in combination with negligible water content suggests that cloud droplets are not present. As a result, the pyroconvection sampled during this penetration is best described as a smoke/ash plume rather

than pyroCu. Additional support for this assertion is provided by Rodriguez et al. (2020) who used temperature and water

vapor profiles from a radiosonde launched at Boise to estimate cloud base at ~6 km MSL, which is several hundred meters

above the flight altitude of penetration 1.

**Flight Level and In Situ Microphysics Data: Penetration 1**



Figure 4. UWKA flight-level and in situ microphysics data for penetration 1 (Fig. 1d, Table 2). (a) Vertical air velocity (blue), (b) temperature (red), relative humidity with respect to liquid (green-solid) and relative humidity with respect to ice (green-dashed), (c) bulk

water content (WC) from the Nevzorov-TWC probe (magenta), Nevzorov-LWC probe (green), LWC-100 probe (gold) and PVM-100 probe (cyan), (d) total concentration ($N_T$) from the 2D-SV probe (blue) and 2D-P probe (red), and (e) area-weighted mean diameter ($\bar{D}_{area}$) from the 2D-SV probe (blue) and 2D-P probe (red). Along-track distance for the penetration shown at bottom.

Total concentrations ($N_T$) from the 2D-SV probe (i.e., particles larger than 10 μm) are almost all greater than zero

during the penetration (Fig. 4d). Maximum values of ~2000 L⁻¹ are evident 224340-224420 UTC and coincident with the

strong updrafts referenced previously. Smaller but still elevated 2D-SV $N_T$ of ~400 L⁻¹ is apparent in the 80 s prior to and

40 s after this period. $N_T$ from the 2D-P probe (i.e., particles larger than 200 μm) is only sporadically greater than zero,

sometimes reaching values of 0.4-0.5 L⁻¹. No particles are being detected by the probe at the other times. Area-weighted

mean diameters ($\bar{D}_{area}$) from the 2D-SV are mostly ~200 μm during the penetration (Fig. 4e). However, there are



occasional spikes of $\overline{D}_{area}$ reaching 500-800 μm. $\overline{D}_{area}$ from the 2D-P is only nonzero at times where the corresponding $N_T$

is nonzero and reaches maximum values of 2000-3000 μm.

The mean particle size distribution over penetration 1 has number concentrations ($\widehat{N}$) that span eight orders of magnitude for diameters 10 μm to 2000 μm (Fig. 5a). Mean $\widehat{N}$ from the 2D-SV is $2.2 \times 10^4$ $L^{-1}$ $mm^{-1}$ at 10 μm diameter while mean $\widehat{N}$ from the 2D-P is $7 \times 10^{-4}$ $L^{-1}$ $mm^{-1}$ at 2000 μm diameter. There is very good overlap between the 2D-SV and CIP measurements of mean $\widehat{N}$. In contrast, the overlap of mean $\widehat{N}$ from the 2D-P with mean $\widehat{N}$ from the 2D-SV and CIP is

relatively poor. The significantly lower concentrations from the 2D-P likely result from the slower electronics of this older probe. To constrain sampling uncertainty, mean $\widehat{N}$ from a given probe is only plotted for size bins that contain at least 10 particles. Without this threshold maximum detected particle diameters from the 2D-SV, CIP and 2D-P extend to 1700 μm, 2300 μm, and 5800 μm, respectively. The number-concentration size distribution of giant ash particles from Fig. 28.6 of Radke et al. (1991) is also plotted in Fig. 5a. These data were collected in a smoke/ash plume generated by a prescribed fire

where the primary fuel was chaparral. There is remarkable agreement between the size distribution from the present study and that from Radke et al. (1991).

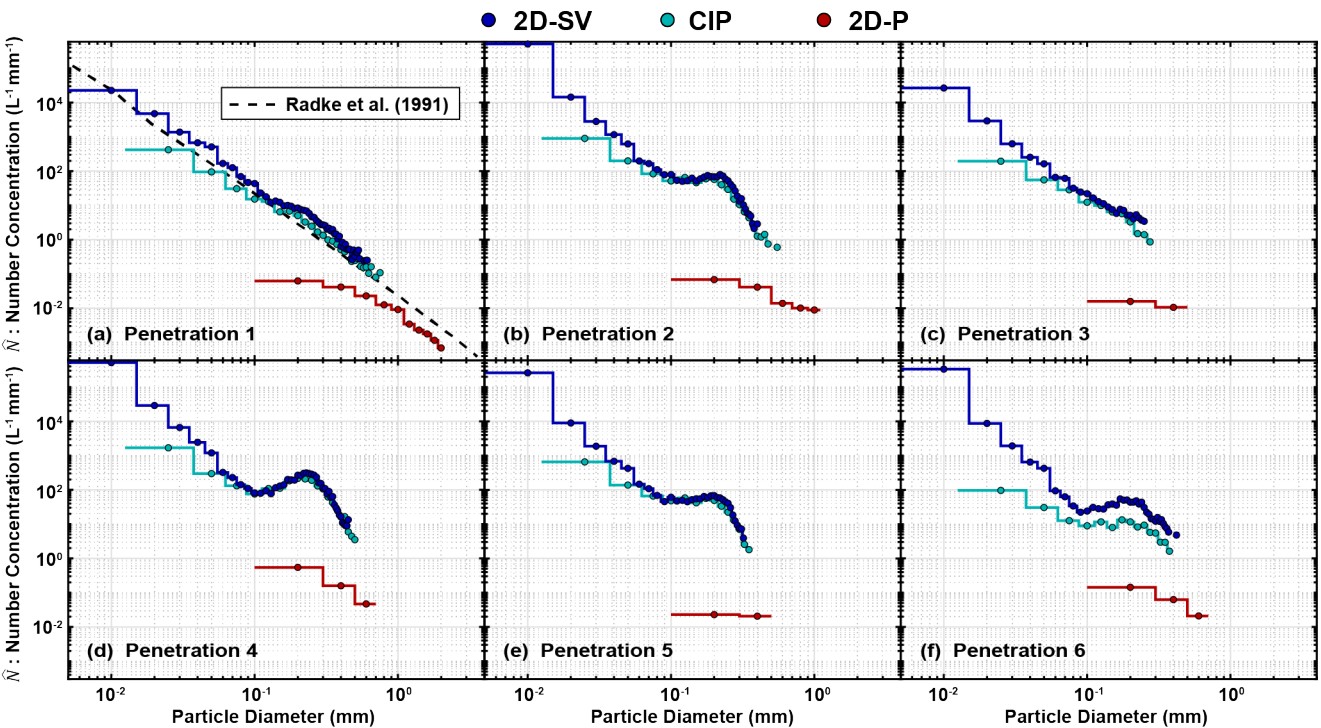

Figure 5. Size distributions of particle number concentration ($\widehat{N}$) for (a-f) penetrations 1-6 (Fig. 1d, Table 2). Data from the 2D-SV, CIP, and 2D-P probes indicated by the blue, cyan, and red filled circles, respectively. Number concentration size distribution of giant ash
particles from Radke et al. (1991) also plotted in (a).

Area concentrations ($\hat{A}$) as a function of diameter averaged over penetration 1 are shaped differently than those for $\hat{N}$ (Fig. 6a). While they extend across the same diameters, variations of $\hat{A}$ span less than four orders of magnitude. Also, the size distribution of $\hat{A}$ is bimodal in nature. This is most apparent in the 2D-SV and CIP data where a secondary peak is evident over the 150-400 μm diameter range. $\hat{A}$ from the 2D-P exhibits a broad peak from 300-1000 mm that may be related

to the secondary peak observed by the 2D-SV and CIP. However, the 2D-P is not sensitive enough to detect the primary peak of the distribution.

**Particle Size Distribtuions: Area Concentration**

● 2D-SV   ● CIP   ● 2D-P

Figure 6. Same as Fig. 5 except for particle area concentration ($\hat{A}$).

A sampling of 2D-SV particle images during penetration 1 indicates a distinct lack of circular symmetry (Fig. 7). Most

particles are characterized by rough edges. Ice crystal aggregates sometimes have this type of appearance, but constituent ice crystal shapes (e.g., needles, dendrites, etc.) are usually apparent to some degree. That is not the case with these images. Flight-level air temperatures of -6° C to -2° C would allow for the existence of ice crystals during the penetration (Fig. 4b). However, values of Nevzorov-TWC are effectively zero (Fig. 4c), indicating that ice particles are likely not present. Also, the updrafts (Fig. 4a) evident when 2D-SV concentrations are highest (Fig. 4d) suggests that these particles originate from

below the aircraft where temperatures are above freezing, which virtually eliminates ice particles as an explanation for the shapes exhibited in Fig. 7. A more reasonable hypothesis is that these are images of pyrometeors, specifically ash particles.



While their sizes are smaller, the shapes of these particles bear a striking resemblance to ash particles created in a burn chamber as shown in Fig. 7 of Baum et al. (2015).

## 2D-SV Images: Penetration 1

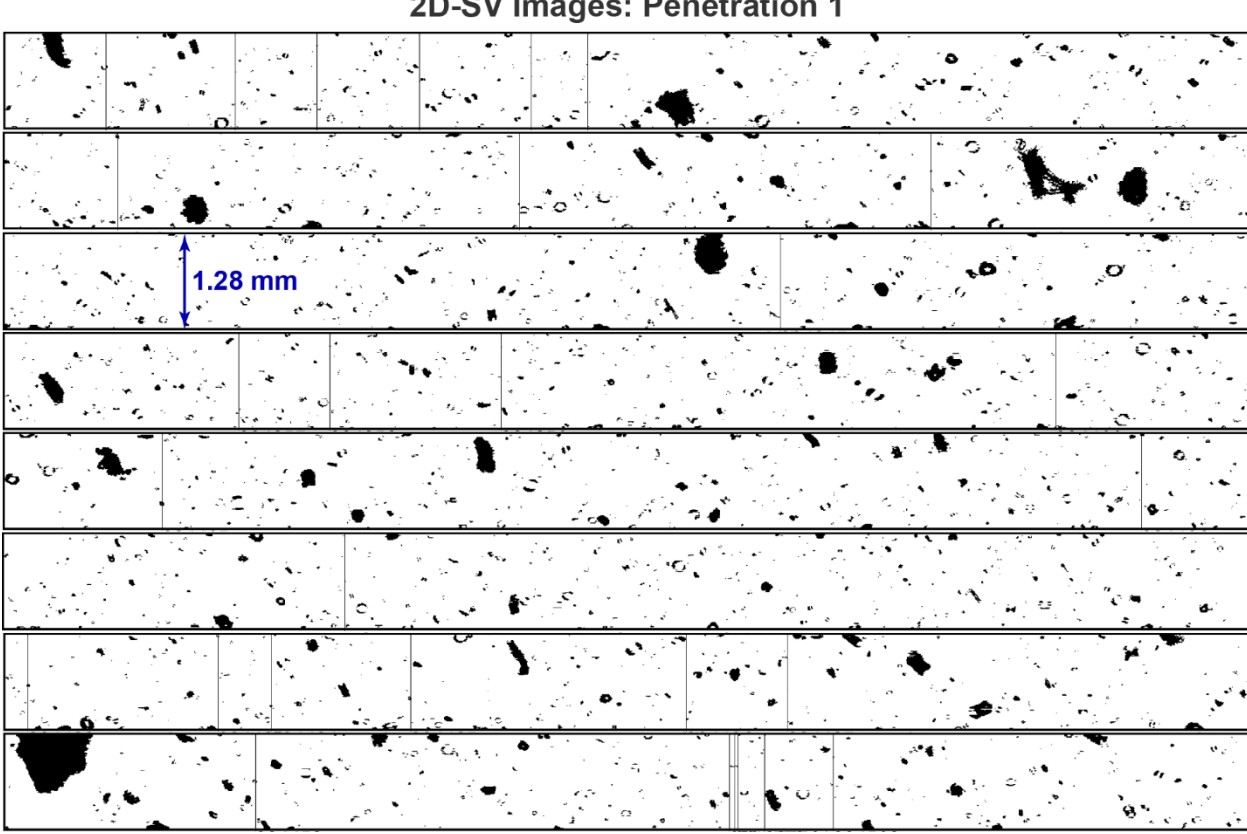

1.28 mm

Figure 7.  Sampling of 2D-SV particle images for penetration 1 (Fig. 1d, Table 2).  Vertical distance of each horizontal strip represents 1.28 mm.

A more quantitative description of 2D-SV particle shapes from penetration 1 is accomplished by analyzing each accepted particle to determine aspect ratio ($ASP_r$), area ($A$), area ratio ($A_r$), and fine-detail ratio ($F_r$).  To bolster data quality, 2D-SV images containing fewer than 25 pixels or that touch one or both diode-array edges are excluded from the particle

shape distribution analysis.  This approach yields 6734 images to quantify particle shape.  The $ASP_r$ distribution has a median of 0.72 and is negatively skewed with values as small as 0.2 (Fig. 8a).  As $ASP_r$ decreases, particles are characterized by more elongated shapes.  The mode and negatively skewed nature of this $ASP_r$ distribution is very similar to that documented in Fig. 6 of Baum et al. (2015).  A dominant mode in the $A$ distribution exists in the smallest area bin (Fig. 9a).  This distribution is characterized by a median of 0.008 mm$^2$ and a long tail that extends to 0.41 mm$^2$.  Baum et al. (2015)

present a distribution of $A$ in their Fig. 3 that is structured similarly (i.e., a dominant mode in the smallest area bin with a long tail) but has values of area that are a factor of ~50 larger due to the relative coarseness of their measurement techniques.  The distribution of $A_r$ is mostly symmetric and centered on a median of 0.48 (Fig. 10a).  This median and the fact that values

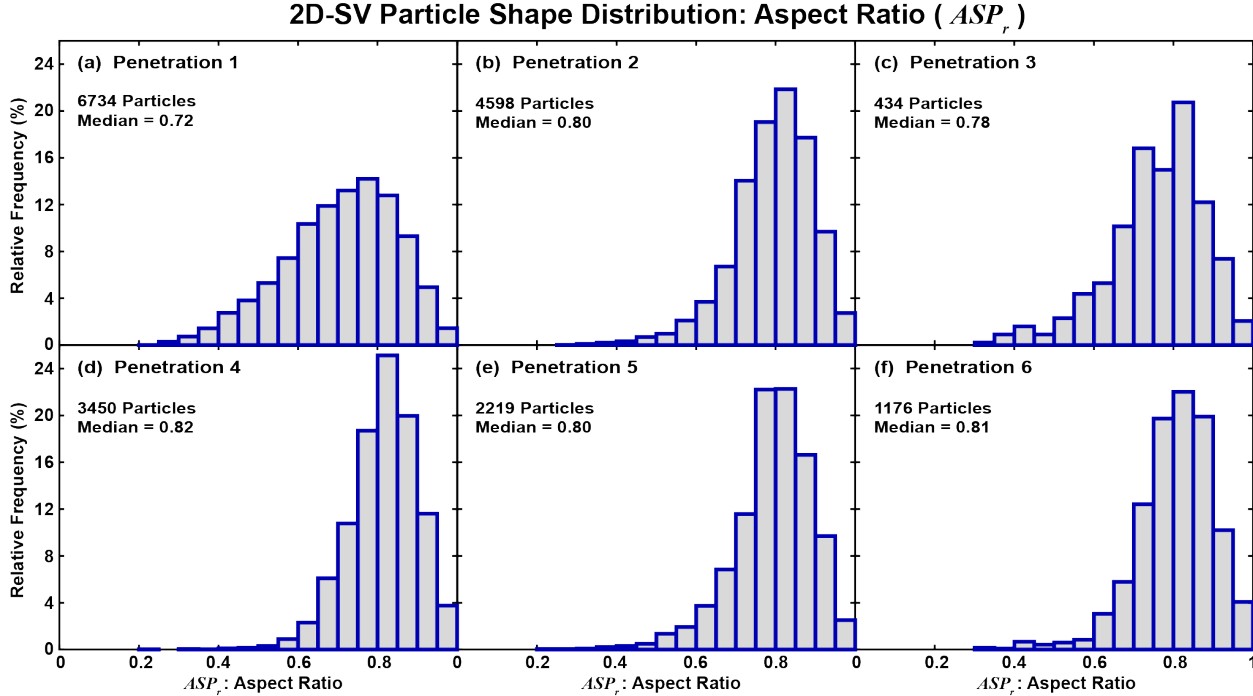

Figure 8.  Histograms of 2D-SV particle aspect ratio ($ASP_r$) for (a-f) penetrations 1-6 (Fig. 1d, Table 2).  Number of particles and distribution median indicated for each penetration.

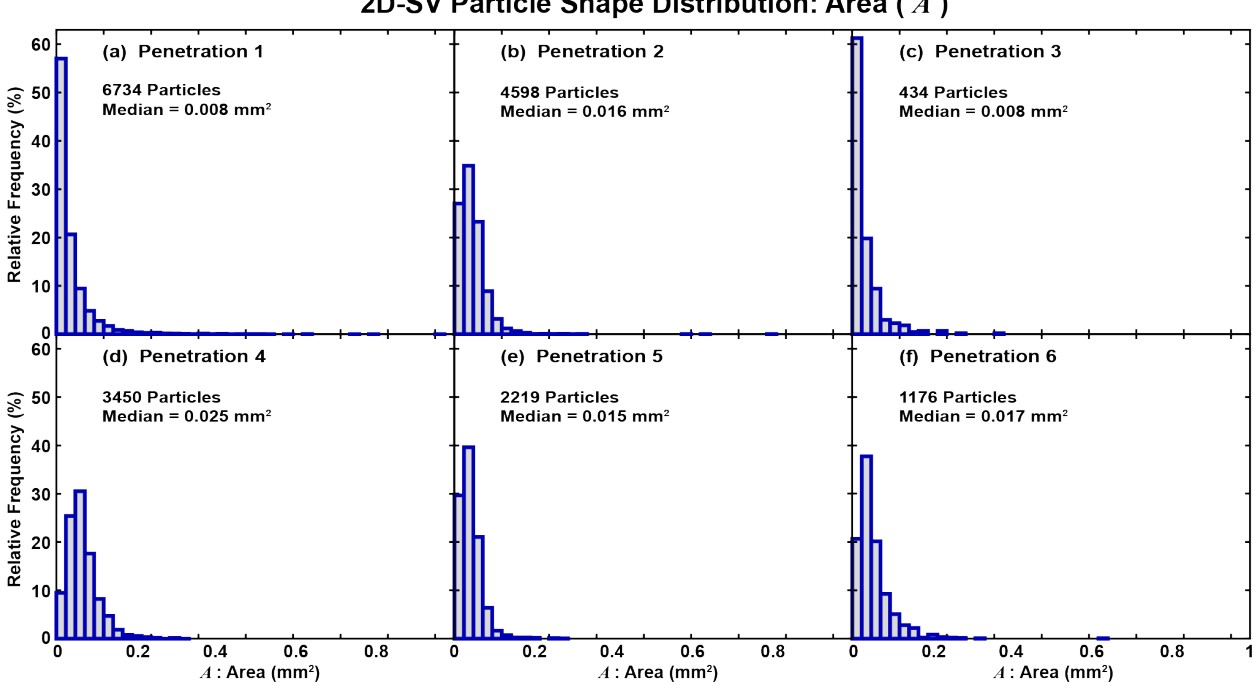

Figure 9.  Same as Fig. 8 except for particle area ($A$).





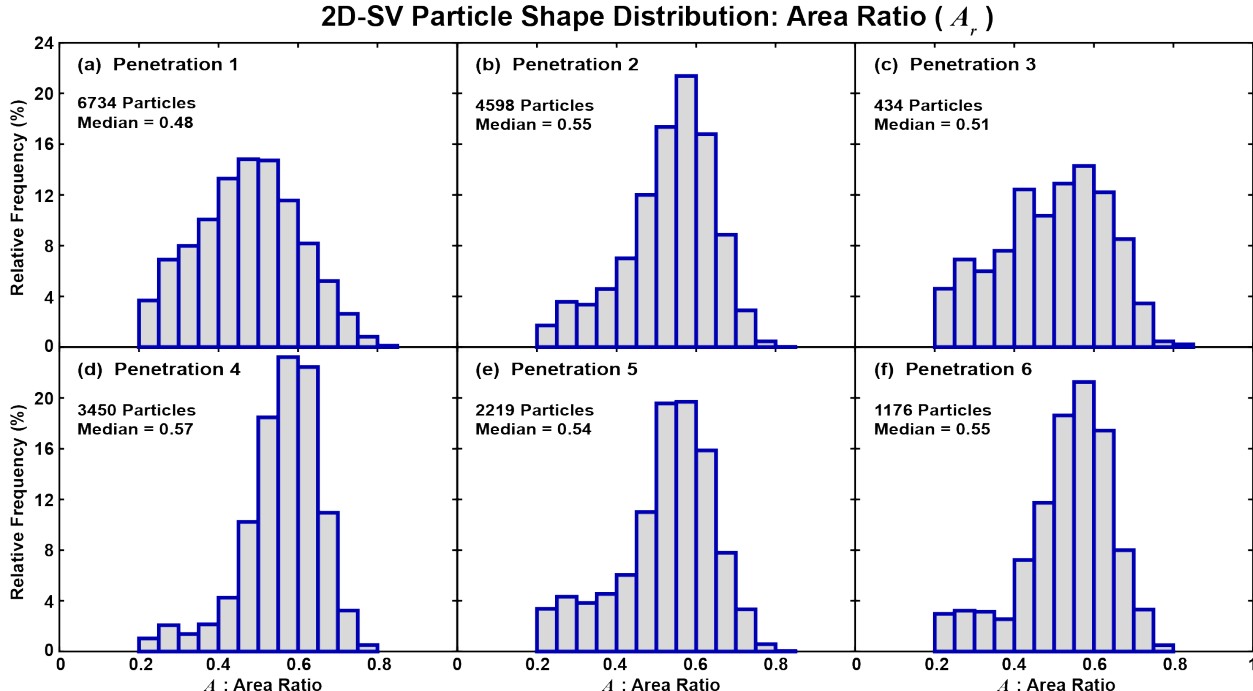

Figure 10. Same as Fig. 8 except for particle area ratio ($A_r$).

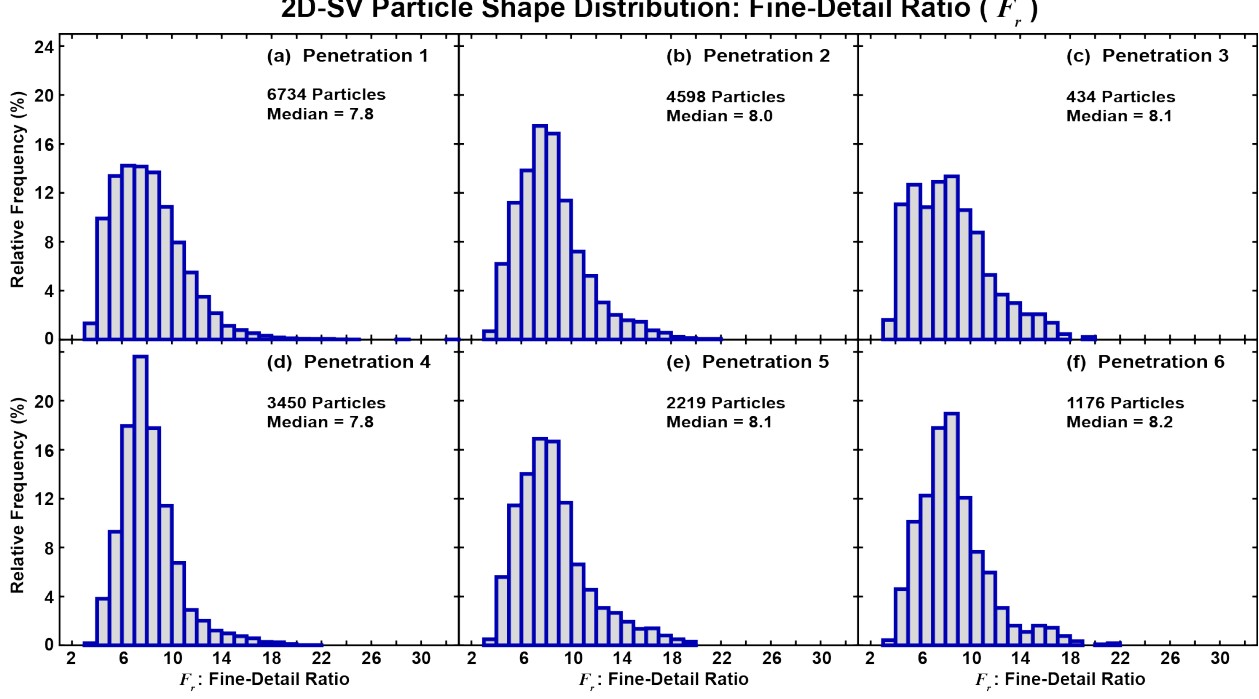


Figure 11. Same as Fig. 8 except for particle fine-detail ratio ($F_r$).





are no larger than 0.85 quantitatively confirm the lack of circular symmetry that is qualitatively evident in the sampling of 2D-SV images (Fig. 7). Many studies have employed $A_r$ to discriminate between different ice particle habits (e.g., Heymsfield and Kajikawa, 1987; Heymsfield and McFarquhar, 1996; Heymsfield et al., 2002). If ice particles were

suspected as being present during penetration 1, the distribution of $A_r$ in Fig. 10a is suggestive of ice crystal aggregates. $F_r$ has also been employed for ice particle habit discrimination (Holroyd, 1987). Values close to 4 are associated with circular shapes while values more than ~13 are associated with particles having considerable porosity (e.g., unrimed aggregates) or significant branching (e.g., dendritic ice crystals). The distribution of $F_r$ has a median of 7.8 and is positively skewed with values as large as 33 (Fig. 11a).

**3.2 Penetrations 2-6: PyroCu**

Penetrations 2-6 are grouped together since, as will be shown, they all sample pyroCus. These penetrations are much shorter in duration (< 60s) and flown at higher levels (7.3-7.7 km MSL) than penetration 1 (Table 2, Fig. 1d). KCBX operates in a clear-air mode, specifically volume coverage pattern 32 (VCP 32; NOAA, 2017), until about 0005 UTC 30 August, which is between penetrations 3 and 4. Thereafter, KCBX switches to VCP 212, which is a less sensitive operating mode optimized

for sampling precipitation. KCBX polarimetric data from VCP 212 looks significantly different than that from VCP 32 collected just before the transition (not shown). As a result of relatively weak reflectivity (< 25 dBZ$_e$) and lower sensitivity, polarimetric data collected during VCP 212 operations is deemed unreliable (e.g., Melnikov and Zrnić, 2007; Ivić et al., 2013). Therefore, only KCBX polarimetric data that accompanies penetrations 2 and 3 is shown in Fig. 12. Penetration 2 (3) is at a range of ~105 km (~100 km) from KCBX, where the 2.5° elevation scan is at ~6.2 km MSL (~5.9 km MSL) and

3.5° elevation scan is at ~8.0 km MSL (~7.7 km MSL). In both penetrations the center of the 3.5° elevation scan is only about 300-400 m above the UWKA flight altitude. The areal extent of reflectivity is smaller at 3.5° (Fig. 12a,g) compared to 2.5° (Fig. 12d,j). Like penetration 1, Z$_{DR}$ is too noisy to infer particle orientation characteristics (Fig. 12b,e,h,k) and ρ$_{hv}$ is small enough (mostly < 0.5) to suggest the presence of irregularly shaped particles (Fig. 12c,f,i,l).

The vertical structure of pyroconvection sampled during penetrations 3-5 is shown in Fig. 13. WCR data is not

available for penetration 2 due to a malfunction and not shown for penetration 6 since the aircraft is turning throughout. The three penetrations supplemented with WCR data indicate passage through the tops of pyroconvective cells confined to the lowest ~5 km AGL (Fig. 13a-c). Lower parts of these cells are tied to the surface in some locations (001655 UTC and 001815 UTC in Fig. 13b and 002555 UTC in Fig. 13c). Reflectivity is also evident above the aircraft in association with penetrations 4-5 (Fig. 13b-c). These echoes are from altostratus and altocumulus clouds unrelated to emissions from the fire.

Maximum reflectivity in association with pyroconvection from penetrations 3-5 is mainly -10 – -5 dBZ$_e$, which is slightly weaker than observed during penetration 1 (Fig. 3a). The exception is reflectivity near the surface at ~001815 UTC immediately after penetration 4 (Fig. 13b) where values are almost 15 dBZ$_e$. These larger values of reflectivity are likely an indication of active combustion at the surface. Updrafts larger than 10 m s$^{-1}$ are common in the pyroconvection sampled





Figure 12. KCBX data approximately coincident with the execution of penetration 2 (a-f) and penetration 3 (g-l). Left column (a,d,g,j) shows reflectivity, middle column (b,e,h,k) shows differential reflectivity ($Z_{DR}$) and right column (c,f,i,l) shows correlation coefficient ($\rho_{hv}$). Colour scales at bottom of each column indicate plotted values. Top row (a-c) is for data at 231633 UTC 29 August 2016 (3.5° elevation angle), second row (d-f) for data at 231503 UTC 29 August 2016 (2.5° elevation angle), third row (g-i) for data at 233603 UTC 29 August 2016 (3.5° elevation angle) and fourth row (j-l) for data at 233433 UTC 29 August 2016 (2.5° elevation angle). Perimeters of the Pioneer Fire at 0600 UTC 29 August 2016 and 0616 UTC 30 August 2016 are indicated with black and red lines, respectively. Black-dashed lines show the locations of penetrations 2 and 3, with green circles and red squares indicating the start and end of each penetration, respectively.





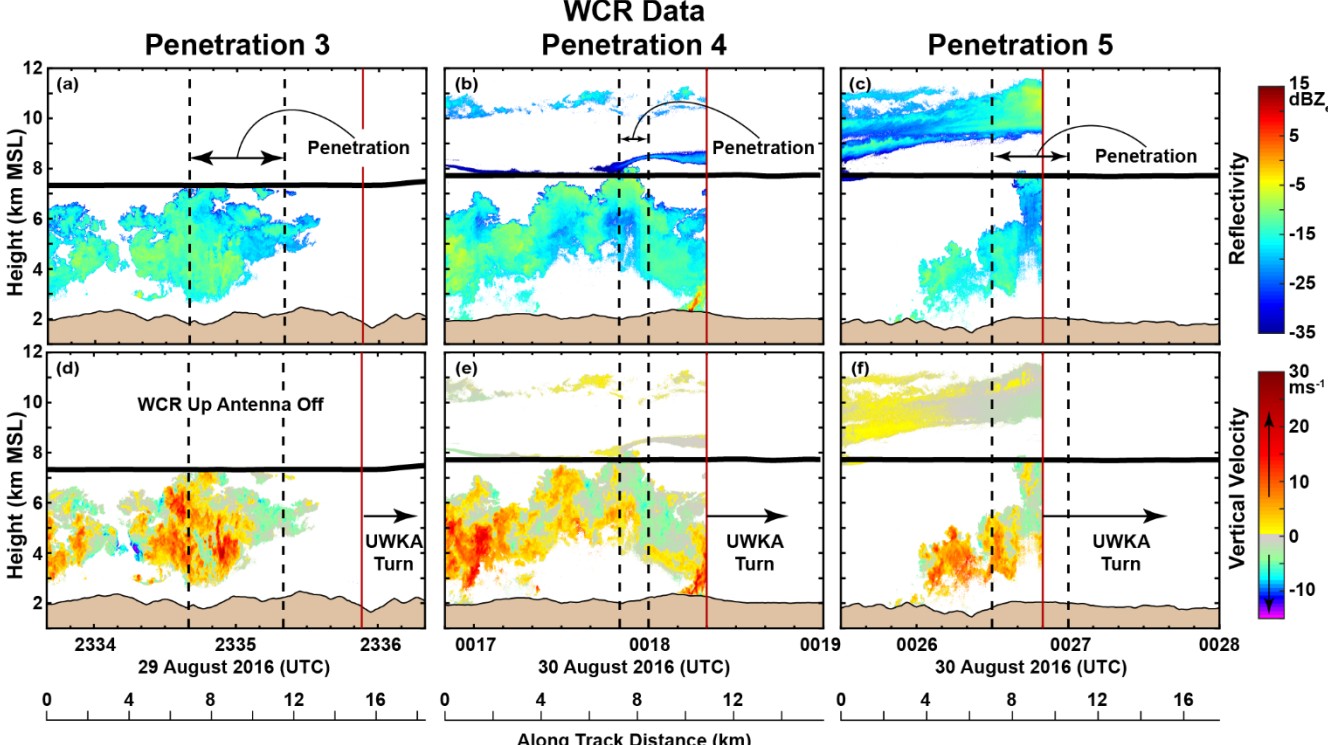

Figure 13. UWKA WCR (a-c) reflectivity and (d-f) radial velocity (effectively vertical velocity) data for (a, d) penetration 3, (b, e) penetration 4, and (c, f) penetration 5 (Fig. 1d, Table 2) and 60 s before and after each of the penetrations (bounded by the vertical, black-dashed lines). Colour scales to the right of each row indicate plotted values. Aircraft track shown by the bold black lines near the middle of each panel. Underlying topography indicated by the tan-filled outline at the bottom of each panel. Along-track distance for the penetrations shown at bottom. Periods without WCR data bounded by vertical red lines.

during penetrations 3-5 (Fig. 13d-f). However, the strongest updrafts, some exceeding 25 m s$^{-1}$, are below flight level, often near the locations where cells are tied to the surface. Downdrafts are smaller in magnitude and spatial scale compared to updrafts, which is like that seen during penetration 1 (Fig. 3b).

Maximum updrafts at flight level are smaller during penetrations 2-6 (Fig. 14a-e) compared to penetration 1 (Fig. 4a). Penetrations 2-3 are characterized by updrafts that are slightly larger than 5 m s$^{-1}$ (Fig. 14a-b), penetrations 4 and 6 by updrafts less than 1 m s$^{-1}$ (Fig. 14c,e), and penetration 5 by no updrafts at all (Fig. 14d). Downdrafts during penetrations 2-6 (Fig. 14a-e) are of comparable magnitude to that observed during penetration 1 (Fig. 4a). Air temperatures across penetrations 2-6 vary between ~-26°C and ~-21°C (Fig. 14f-j), which is significantly colder than penetration 1 (Fig. 4b) due to the higher flight altitudes. Interestingly, there are relatively cold temperatures in the updraft core of penetration 2 (Fig. 14f), which may be a signature of pyroconvection overshooting its equilibrium level. Maximum values of relative humidity are generally larger during penetrations 2-6 compared to penetration 1, with values exceeding 85% during penetrations 2-5 (Fig. 14f-i). Significant values of bulk water content are present in penetrations 2-6 (Fig. 14k-o), a stark contrast with the negligible values observed in penetration 1. The substantial cloud liquid water content observed during penetrations 2-6





suggests that this pyroconvection is best described as pyroCu. Measurements from the PVM-100 probe produce consistently larger values of cloud liquid water content (~0.6 g m$^{-3}$ to ~1.2 g m$^{-3}$) compared to those from the Nevzorov-LWC and LWC-100 probes (~0.2 g m$^{-3}$ to ~0.8 g m$^{-3}$). Notably, values of total condensed water content from the Nevzorov-TWC probe are

always lower than values of cloud liquid water content from the Nevzorov-LWC probe. This artifact is likely due to the lower collection efficiency of the Nevzorov-TWC probe compared to the Nevzorov-LWC probe for cloud droplets less than 20 μm in diameter (Korolev et al., 1998; Schwarzenboeck et al., 2009). Data from the PVM-100 probe indicate cloud-droplet effective radii ($R_e$) of ~5 μm during penetrations 2-6 (Fig. 14k-o).

**Flight Level and In Situ Microphysics Data**

Figure 14. UWKA flight-level and in situ microphysics data for penetrations 2-6 (Fig. 1d, Table 2). (a-e) Vertical air velocity (blue), (f-j) temperature (red), relative humidity with respect to liquid (green-solid) and relative humidity with respect to ice (green-dashed), (k-o) bulk water content (WC) from the Nevzorov-TWC probe (magenta), Nevzorov-LWC probe (green), LWC-100 probe (gold), PVM-100 probe (cyan), and cloud-droplet effective radii ($R_e$) from the PVM-100 probe (blue), (p-t) total concentration ($N_T$) from the 2D-SV probe (blue) and 2D-P probe (red), and (u-y) area-weighted mean diameter ($\overline{D}_{area}$) from the 2D-SV probe (blue) and 2D-P probe (red). Along-track distance for each penetration shown at bottom.



Particle characteristics as measured by the 2D-SV during penetrations 2 and 4-6 (Fig. 14p,r-t,u,w-y) differ substantially from the smoke/ash plume sampled during penetration 1 (Fig. 4d-e). Specifically, maximum values of 2D-SV $N_T$ are about one order of magnitude larger (Fig. 14p,r-t) and maximum values of 2D-SV $\overline{D}_{area}$ are smaller by a factor of ~3-5 (Fig.

14u,w-y). This tendency may be due to high concentrations of small cloud droplets. The outlier in this trend is penetration 3, where maximum values of 2D-SV $N_T$ and $\overline{D}_{area}$ (Fig. 14q,v) are comparable to those observed during penetration 1 (Fig. 4d-e). Differences are also evident in 2D-P particle characteristics. Maximum 2D-P $\overline{D}_{area}$ is smaller for penetrations 2-6 (Fig. 14u-y) compared to penetration 1 (Fig. 4e), but only by a factor of ~2-4. Maximum 2D-P $N_T$ is slightly smaller for the pyroCu penetrations (Fig. 14p-t vs Fig. 4d), perhaps owing to the smaller values of maximum 2D-P $\overline{D}_{area}$ and that the 2D-P

is not able to detect small cloud droplets.

The mean particle size distribution of $\widehat{N}$ over penetration 3 (Fig. 5c) is like that observed in penetration 1 (Fig. 5a) for 2D-SV and CIP measurements up to ~300 µm in diameter. However, unlike penetration 1, particles larger than 400 µm are not observed, even by the 2D-P. The other pyroCu penetrations are all characterized by considerably different size distributions of $\widehat{N}$ (Fig. 5b,d-f) compared to penetration 1 (Fig. 5a). Specifically, mean $\widehat{N}$ from the 2D-SV and CIP is larger

in penetrations 2 and 4-6 by more than an order of magnitude over the 10-500 µm diameter range (Fig. 5b,d-f), especially at smaller diameters. Particles larger than 500 µm are detected in low concentrations by the 2D-P, but only up to 600-1000 µm, which is smaller than observed during penetration 1 (Fig. 5a). The size distributions of $\widehat{N}$ for penetrations 2 and 4-6 are also characterized by a bimodal structure, with secondary peaks in the 200-300 µm diameter range (Fig. 5b,d-f). This type of bimodal structure is not evident in penetration 1 (Fig. 5a).

Mean particle size distributions of $\widehat{N}$ over penetrations 2-6 (Fig. 5b-f) are significantly different from those documented for pyroCb in Fig. 11 of Peterson et al. (2022). One difference is that the particle number concentrations detected in Peterson et al. (2022) are generally larger than those observed in the present study. Also, the Peterson et al. (2022) study observes particles that are larger than 1 mm whereas almost all the pyroCu particles in the present study are less than 1 mm. These differences are perhaps not surprising given that the pyroCb in Peterson et al. (2022) are associated with ground-based

NWS radar reflectivity larger than 40 dBZ$_e$, likely indicative of substantial hydrometeor generation in the deep clouds they sampled from -40°C to -18°C. In contrast, KCBX radar reflectivity from the present study maximizes at ~25 dBZ$_e$ (Figs. 1d, 12a,d,g,j).

Size distributions of $\widehat{A}$ are bimodal for penetrations 2 and 4-6 (Fig. 6b,d-f). While this structure is also evident in penetration 1 (Fig. 6a), it is considerably more pronounced during penetrations 2 and 4-6, with mean $\widehat{A}$ 1-2 orders of

magnitude larger over the 150-400 µm diameter range. Interestingly, the size distribution of $\widehat{A}$ during penetration 3 (Fig. 6c) is like penetration 1 (Fig. 6a) up to ~100 µm, but at larger diameters a bimodal structure is not evident (Fig. 6c).

2D-SV images sampled during penetrations 2-6 (Fig. 15) are characterized by a relatively high ratio of small (diameter < 50 µm) particles whose shape is ill defined due to the resolution of the probe. Given the presence of substantial cloud liquid water content (Fig. 14k-o), these small particles are likely cloud droplets. Such small particles are not evident in





the 2D-SV imagery from penetration 1 (Fig. 7). For particles larger than ~50 μm, 2D-SV imagery from penetrations 2-6 (Fig. 15) qualitatively suggests a somewhat higher degree of circular symmetry than observed during penetration 1. However, many of these larger particles are still characterized by rough edges.

Quantitative analysis of 2D-SV images from penetrations 2-6 provides a more robust description of particle shapes and how they compare with those from penetration 1. After excluding images containing fewer than 25 pixels or that touch one

of both edge diodes, penetrations 2-6 employ 4598, 434, 3450, 2219, and 1176 images, respectively, to quantify particle shape. The $ASP_r$ distributions for penetrations 2-6 (Fig. 8b-f) are narrower and have larger medians than observed during penetration 1 (Fig. 8a). Penetration 3 exhibits an $A$ distribution with a mode and median like penetration 1 (Fig. 9c vs Fig. 9a). In contrast, penetrations 2 and 4-6 have $A$ distributions with larger medians and modes that exist in the second or third area bins (Fig. 9b,d-f). Fewer particles with $A$ greater than 0.1 mm$^2$ are evident in penetrations 2-6 (Fig. 9b-f) than in

penetration 1 (Fig. 9a). The $A_r$ distribution of penetration 3 is also like that of penetration 1 (Fig. 10c vs Fig. 10a) in terms of breadth, median, and relative symmetry about the median. Penetrations 2 and 4-6 are characterized by narrower and negatively skewed distributions of $A_r$ with larger medians (Fig. 10b,d-f). These larger values of $A_r$ coupled with the larger values of $ASP_r$ in penetrations 2-6 are consistent with the qualitative observation of greater circular symmetry in particles sampled during the pyroCu penetrations (Fig. 15) compared to penetration 1 (Fig. 7). The distributions of $F_r$ from

penetrations 2-6 (Fig. 11b-f) are generally like that from penetration 1 (Fig. 11a).

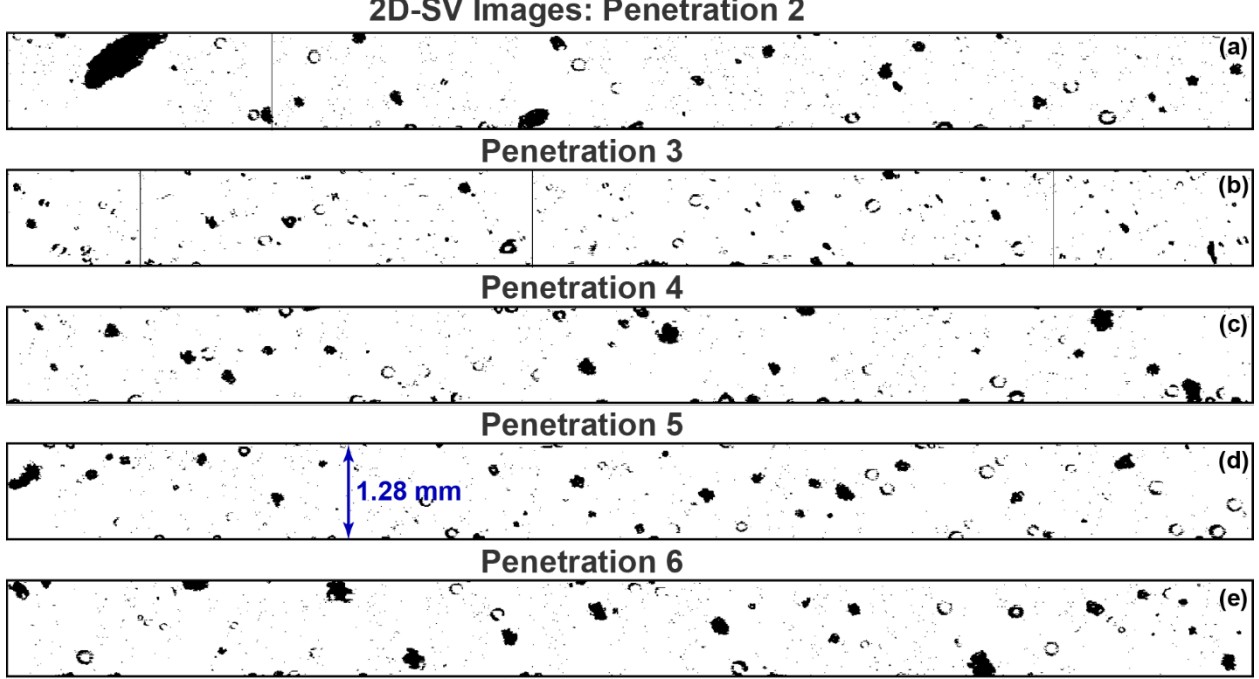

Figure 15. Sampling of 2D-SV particle images for penetrations 2-6 (Fig. 1d, Table 2). Vertical distance of each horizontal strip represents 1.28 mm.



While particles less than 50 μm in diameter during penetrations 2-6 are likely cloud droplets, the nature of larger

particles is unclear since ice particles, pyrometeors and a combination of the two are plausible explanations. Pyrometeors in the form of ash particles could be lofted upward to the tops of the pyroCus, especially in penetrations 2-3 where updrafts larger than 1 m s$^{-1}$ are observed (Fig. 14a-b). Likewise, ice particles are possible since air temperatures at flight level are well below freezing (Fig. 14f-j). Also, conditions of ice supersaturation are evident in four of the five penetrations (Fig. 14f-i). Finally, the distributions of $A_r$ and $F_r$ in Figs. 10b-f and 11b-f, respectively, are consistent with values expected

for ice particles. If these larger particles are composed of ice, they should produce a signal in the Nevzorov-TWC data whereas ash particles should not be detectable by the Nevzorov probe. This analysis is complicated by the presence of small cloud droplets that are under-sampled by the Nevzorov-TWC probe relative to the Nevzorov-LWC probe. The collection efficiency of ~5 μm radii cloud droplets (Fig. 14k-o) by the Nevzorov-LWC probe is ~0.98 while that for the Nevzorov-TWC probe is ~0.63±0.09 (Fig. 4 of Korolev et al., 1998). For a volume only containing cloud droplets, the ratio of TWC

from the Nevzorov-TWC probe to LWC from the Nevzorov-LWC probe ($NEV_r = NEV_{TWC}/NEV_{LWC}$) should reflect the ratio of the corresponding collection efficiencies. However, values of $NEV_r$ are not constant during penetrations 2-6 (Fig. 16). The smallest $NEV_r$ occurs during penetration 3 (Fig. 16b) and is ~0.4. This suggests a smaller collection efficiency than expected from Korolev et al. (1998), which may be due to an overestimate of cloud droplet effective radius by the PVM-100 probe. The largest $NEV_r$ values of ~0.7 occur during penetrations 4 and 6 (Fig. 16c,e).

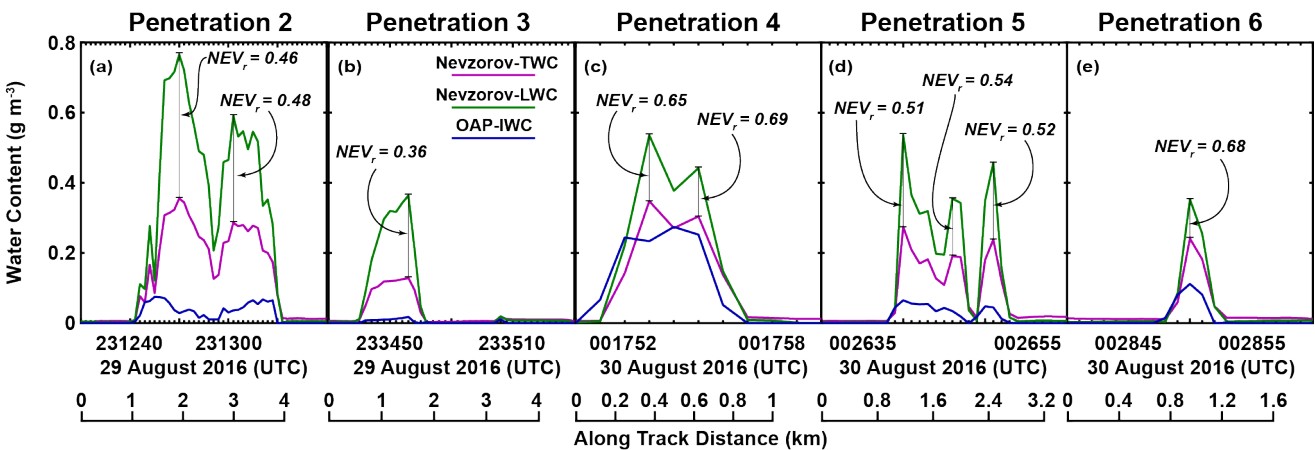


Figure 16. UWKA water content data for (a-e) penetrations 2-6 (Fig. 1d, Table 2). TWC from the Nevzorov-TWC probe (magenta), LWC from the Nevzorov-LWC probe (green), and IWC derived from OAP size distributions using a mass-diameter relation (blue). Ratio of TWC from Nevzorov-TWC probe to LWC from Nevzorov-LWC probe ($NEV_r$) shown at selected times. Along-track distance for each penetration shown at bottom.

If cloud droplet radii are assumed to be the same for each of the penetrations, then variations of $NEV_r$ might be explained by the presence of ice particles. An independent measure of IWC is needed to test this hypothesis. Size distributions of OAP number concentration at 1 Hz are integrated over diameter with a mass-diameter relationship for ice-crystal aggregates from Brown and Francis (1995) to derive IWC ($OAP_{IWC}$). The derivation is restricted to particles having





diameters greater than or equal to 50 μm to eliminate the contribution of cloud droplets. Results of this analysis show that

the smallest $OAP_{IWC}$ occurs during penetration 3, which coincides with the smallest $NEV_r$ (Fig. 16b). The largest and next largest $OAP_{IWC}$ are observed in penetrations 4 and 6, respectively where $NEV_r$ is largest (Fig. 16c,e). These trends suggest that ice particles larger than 50 μm diameter are present in the data produced by the Nevzorov probe and OAP's. However, this analysis does not eliminate the possibility that pyrometeors are also in the sample volume. Indeed, many of the measurements from penetration 3 show more similarity to those for the smoke/ash plume in penetration 1 than those for the

pyroCu in penetrations 2 and 4-6. This suggests that the particles sampled during penetration 3 might be composed of a greater proportion of pyrometeors than the particles sampled during the other pyroCu penetrations.

## 4 Summary and conclusions

This study has characterized the size and shape distributions of 10 μm to 6 mm diameter particles sampled during penetrations of pyroconvection by the UWKA research aircraft over the period 29-30 August 2016. The pyroconvection, in

the form of both smoke/ash plumes and pyroCus, was produced by the Pioneer Fire, a large wildfire (ultimately 76,081 ha) located northeast of Boise, Idaho (Fig. 1). In situ measurements by the UWKA included standard flight level parameters (e.g., navigation, winds, atmospheric state), bulk water content and particle concentration, size, and shape. Additionally, airborne Doppler radar observations from the WCR were used to characterize the depth and vertical motions of the sampled pyroconvection.

Penetration 1 probed a smoke/ash plume at 5.2 km MSL characterized by strong updrafts, many larger than 10 m s$^{-1}$ and one of almost 36 m s$^{-1}$ (Figs. 3b, 4a). This penetration spanned a temperature range of -6°C to -2°C, was subsaturated with relative humidity less than 80%, and contained negligible amounts of bulk water content (Fig. 4b,c). The size distribution of number concentration was very similar to that documented by Radke et al. (1991) for a smoke/ash plume from a prescribed fire (Fig. 5a). Also, particle shapes exhibited qualitative and quantitative attributes (Figs. 7, 8a, 9a) comparable

to ash particles created in a burn chamber by Baum et al. (2015). These comparisons support the conclusion that particles sampled during penetration 1 were most likely pyrometeors composed of ash, an assertion bolstered by the argument that these particles originated from below the aircraft where above freezing temperatures virtually eliminates ice particles as an explanation for their composition.

    PyroCus were sampled during penetrations 2-6 at 7.3-7.7 km MSL. While there were some strong updrafts larger than

10 m s$^{-1}$ at lower levels (Fig. 13d-f), updrafts at flight level (Fig. 14a-e) were weaker than those observed during penetration 1. Temperatures were colder, spanning a range of -26°C to -21°C and relative humidity was generally larger, often exceeding 85% and sometimes nearing saturation with respect to liquid (Fig. 14f-j). Measured values of cloud liquid water content (Fig. 14k-o) were also significantly larger than observed during penetration 1. Size distributions of number concentration in penetrations 2 and 4-6 (Fig. 5b,d-f) were characterized by considerably larger concentrations than

penetration 1. Also, the size distributions in penetrations 2 and 4-6 exhibited bimodal structures, with a secondary a peak at



200-300 μm. This bimodal structure was not evident in penetration 1. Qualitative and quantitative analysis of particle shapes for penetrations 2-6 (Figs. 15, 8b-f, 10b-f) suggested a somewhat higher degree of circular symmetry compared to penetration 1. Particle composition in the pyroCu penetrations was more ambiguous than in the smoke/ash plume penetration where pyrometeors were likely present. Evidence was presented to suggest that hydrometeors in the form of ice

particles were sampled in the pyroCu penetrations (Fig. 16). However, the joint existence of pyrometeors in the form of ash particles was not eliminated as a possibility, particularly for penetration 3.

The in situ observations documented in this study contribute to our knowledge of wildfire-induced pyroconvection particles larger than smoke particulates and cloud droplets, but the sample size is still relatively small. Additional in situ observations are needed across a broad spectrum of pyroconvective features from smoke/ash plumes to pyroCus to pyroCbs.

These in situ observations should occur in concert with observations that can document vertical motions in the pyroconvection and fire characteristics at the surface, including fire radiative power. Obtaining such in situ observations is not a trivial matter given the highly turbulent nature of pyroconvection and safety concerns in operating aircraft in those environments. Efforts should be undertaken to identify and develop research aircraft that can meet those challenges, whether they are crewed or uncrewed. In the absence of such observations the community will be reliant on remote-sensing

observations, such as from ground-based polarimetric radars, to make inferences about the characteristics of pyroconvection particles. While these remote-sensing observations can be useful, they have not been adequately calibrated for pyroconvection through validation with in situ observations.


*Data availability*.

The UWKA data used in this study are available at https://www.uwyo.edu/atsc/uwka/facility-data-requests.html. The KCBX radar data used in this study are available at https://www.ncdc.noaa.gov/nexradinv/chooseday.jsp?id=kcbx.

*Author contributions*.

**D. E. Kingsmill**: Conceptualization; data curation; formal analysis; funding acquisition; methodology; project administration; software; visualization; writing-original draft; writing-review and editing.

**J. R. French**: Data curation; formal analysis; software; writing-review and editing.

**N. P. Lareau**: Conceptualization; data curation; writing-review and editing.


*Competing interests*.

The authors declare that they have no conflicts of interest.



*Acknowledgements.*

We acknowledge and are thankful for the efforts of staff at the University of Wyoming King Air Research Aircraft Facility in planning and executing the aircraft mission described in this study as well as processing the data that was collected. Craig B. Clements of San José State University was a vital collaborator in securing funding to support the aircraft mission. Discussions with Sandra E. Yuter of North Carolina State University assisted in the interpretation of KCBX polarimetric

radar data. This research is supported under grant AGS-1719243 from the National Science Foundation.

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
