# Peer review of "In Situ Microphysics Observations of Intense Pyroconvection from a Large Wildfire"

_Atmospheric Chemistry and Physics, 2022_

## Author Comment (AC1)

Atmos. Chem. Phys. Discuss., referee comment RC1
https://doi.org/10.5194/acp-2022-610-RC1, 2022
**Comment on acp-2022-610**

Anonymous Referee #1

Referee comment on "In Situ Microphysics Observations of Intense Pyroconvection from a Large Wildfire" by David E. Kingsmill et al., Atmos. Chem. Phys. Discuss., https://doi.org/10.5194/acp-2022-610-RC1, 2022

This paper describes in situ microphysics observations of intense pyroconvection for a large wildfire in Idaho from an airborne platform. Content includes both dry (mainly smoke/ash) and moist (pyroCu) pyroconvective plumes. A variety of parameters are presented to showcase microphysics data for the diameter range of 10 μm to 6 mm. This is supplemented with weather instrumentation and a cloud radar onboard the aircraft, along with ground-based weather radar. Results from this study fill a critical gap in measurements of pyroconvection, especially for pyroCu. The content is well organized, the figures are of high quality, and the narrative is generally easy to follow. I recommend publication after addressing the minor edits below.

The hardest part for me to follow was in the last few paragraphs on the pyroCu discussion (Lines ~460-490). The fine details of the Nevzorov probe clutter the messaging about cloud droplets, cloud ice vs. pyrometeors. Perhaps some rearrangement of the sentences might help. Think of readers less familiar with the details of the instruments, who want to know what's going on inside a pyroCu.

The last two paragraphs of section 3.2 have been significantly revised to increase clarity. Below is the revised text:

"While particles less than 50 μm in diameter during penetrations 2-6 are likely cloud droplets, the nature of larger particles is unclear since ice particles, pyrometeors and a combination of the two are plausible explanations. Pyrometeors in the form of ash particles could be lofted upward to the tops of the pyroCus, especially in penetrations 2-3 where updrafts larger than 1 m s$^{-1}$ are observed (Fig. 14a-b). Likewise, ice particles are possible since air temperatures at flight level are well below freezing (Fig. 14f-j). Also, conditions of ice supersaturation are evident in four of the five penetrations (Fig. 14f-i). Finally, the distributions of $A_r$ and $F_r$ in Figs. 10b-f and 11b-f, respectively, are consistent with values expected for ice particles. If these larger particles are composed of ice, they should produce a signal in the

Nevzorov-TWC ($NEV_{TWC}$) data whereas ash particles should not be detectable by the Nevzorov probe. However, this analysis is complicated by the presence of small cloud droplets. Cloud droplets also produce a signal in the $NEV_{TWC}$ data, but the magnitude of that signal depends not only on the water content, but also on the size of the droplets themselves. For the same liquid water contents, the signal in $NEV_{TWC}$ will increase by ~50% as droplet effective radii increase from ~5 to ~15 μm, while over the same range of radii, the signal in $NEV_{LWC}$ only increases by ~2% (Fig. 4 and Equation 13 of Korolev et al., 1998). Thus, clouds composed of liquid, even when devoid of ice, will have different signals in $NEV_{TWC}$ and $NEV_{LWC}$ depending on droplet size regardless of liquid water content.

Values of $NEV_{TWC}$, $NEV_{LWC}$ and the ratio of these signals ($NEV_r = NEV_{TWC}/NEV_{LWC}$) are presented for penetrations 2 through 6 in Fig. 16. Following the discussion in the previous paragraph and the definition of $NEV_r$, an increase in $NEV_r$ may result from either (or both) an increase in droplet effective radius and/or an increase in IWC. In penetration 3, $NEV_r$ is 0.36 (Fig. 16b). Assuming the cloud is devoid of ice, this value corresponds to a droplet effective radius of ~2.5 μm. If, on the other hand, ice was present, then the radius would be even smaller. This radius is about half of that measured by the PVM-100 probe across all penetrations (Fig. 14 k-o) suggesting that the PVM may be overestimating the droplet effective radius by a factor of 2. $NEV_r$ varies from 0.36 to 0.69 across all five penetrations. If cloud droplet effective radii are assumed to be the same across these penetrations, then the variations in $NEV_r$ might be explained by the presence of ice particles. To further investigate the possible presence of ice, an independent measure of IWC is needed. Size distributions of OAP number concentration at 1 Hz are integrated over diameter with a mass-diameter relationship for ice-crystal aggregates from Brown and Francis (1995) to derive IWC ($OAP_{IWC}$). The derivation is restricted to particles having diameters greater than or equal to 50 μm to eliminate the contribution of cloud droplets. Results of this analysis show that the smallest $OAP_{IWC}$ occurs during penetration 3, which coincides with the smallest $NEV_r$ (Fig. 16b). The largest and next largest $OAP_{IWC}$ are observed in penetrations 4 and 6, respectively where $NEV_r$ is largest (Fig. 16c,e). These trends suggest that ice particles larger than 50 μm diameter are present in the data produced by the Nevzorov probe and OAP's. However, this analysis does not eliminate the possibility that pyrometeors are also in the sample volume. Indeed, many of the measurements from penetration 3 show more similarity to those for the smoke/ash plume in penetration 1 than those for the pyroCu in penetrations 2 and 4-6. This suggests that the particles sampled during penetration 3 might be composed of a greater proportion of pyrometeors than the particles sampled during the other pyroCu penetrations."

The differences for penetration #3 compared with the other pyroCu data is interesting. Might the biomass/vegetation be different in that part of the fire front? It's the only penetration along the eastern part of the fire front. Perhaps being a bit lower in altitude was enough of a factor?

There is no evidence to indicate that the biomass/vegetation was significantly different for penetration 3 compared to the other pyroCu penetrations. Also, penetration 3 is immediately adjacent to penetration 2 and separated by only a few km. As you indicate, the altitude of penetration 3 is a little lower (7.3 km MSL) than the other pyroCu penetrations (7.7 km MSL). However, 400 m lower does not seem very significant in comparison to the 2.1 km difference in altitude between penetrations 3 and 1. We have added some text to the end of section 3.2 to address this issue:

"This suggests that the particles sampled during penetration 3 might be composed of a greater proportion of pyrometeors than the particles sampled during the other pyroCu penetrations. The reasons for this difference are not clear. Penetration 3 is executed at 7.3 km MSL, about 400 m lower than penetrations 2 and 4-6 but still 2.1 km higher than penetration 1 (Table 2). Also, there is no evidence to indicate that underlying vegetation associated with penetration 3 is significantly different than that for penetrations 2 and 4-6."

Is there anything you can say in the conclusions on how these data might be used in fire-scale modeling work?

We feel that these data are potentially useful to investigators that employ numerical models to simulate wildfires and related pyroconvection. Specifically, these data might be useful in validating and improving those models. However, we are not well positioned to advise those investigators exactly how they should use the data. We have modified the beginning of the last paragraph of the conclusions to address this issue:

"The in situ observations documented in this study contribute to our knowledge of wildfire-induced pyroconvection particles larger than smoke particulates and cloud droplets. These data could be used to validate and improve models that simulate wildfires and related pyroconvection. They also could be employed to advance the application of radar to study pyroconvection by relating particle size distributions to reflectivity values. However, the sample size is still relatively small."

Anything you can say on the potential for precipitation development should these pyroCu continue developing into a pyroCb?

It is not clear to us, but we think you are asking why the effectively precipitation-free pyroCu we sampled did not develop into a pyroCb with clear evidence of precipitation. If so, this seems like a very open-ended question with many possible dimensions. The updrafts in this pyroCu as documented in Rodriguez et al. (2020) and our study are significant (up to ~60 m/s), which are clearly large enough to condense liquid water. However, that condensate is not able to undergo a conversion to larger-sized (~> 1 mm) liquid and/or ice hydrometeors that could be reasonably described as precipitation from an in situ or radar observation perspective. Unfortunately, we do not have the microphysical observations to credibly explore this issue. For example, it is possible that further downstream the microphysics evolved in the upper plume (e.g., the higher echo tops to the NE in Fig. 1b), but no flight data were obtained in that critical region. As such, anything that we could add to the narrative would be highly speculative, which we believe would detract from the focus of the study as a whole. As a result, we are hesitant to address this issue in the manuscript, though hope to have observations in the future to address these questions.

Are there any existing observations of traditional cumulus clouds for a direct microphysics

comparison with the pyroCu? Ideally, this would be in a similar thermodynamic environment.

There are numerous microphysical observations of traditional (i.e., non-pyro) cumulus clouds. However, what characteristics would these traditional cumulus clouds need to possess to allow a fair comparison? You mention that the thermodynamic environment should be similar. We agree. But what about other characteristics such as cloud depth and vertical motions? Traditional cumulus clouds with the depth and vertical motions of the pyroCu sampled in our study are associated with supercell thunderstorms. We are not aware of any in situ microphysics observations from the interior of a supercell thunderstorm. Therefore, we do not feel that a fair comparison is possible. It is our hope that the data in our manuscript, and similar observations in Peterson et al. 2022 (BAMS), will make comparisons amongst different pyroCu/Cb cases and relative to Cu/Cb cases possible in the future when we will hopefully have more cases to compare.

You may consider making the figure letters a bit larger for some of the panels.

The minimum font size on almost all figures has been increased.

---

## Author Comment (AC2)

Atmos. Chem. Phys. Discuss., referee comment RC2
https://doi.org/10.5194/acp-2022-610-RC2, 2022
**Comment on acp-2022-610**

Anonymous Referee #2

Referee comment on "In Situ Microphysics Observations of Intense Pyroconvection from a Large Wildfire" by David E. Kingsmill et al., Atmos. Chem. Phys. Discuss., https://doi.org/10.5194/acp-2022-610-RC2, 2022

**Overview**

This study evaluated airborne in situ and remote sensing and ground-based remote sensing measurements of wildfire-induced pyroconvection. The authors analyzed the shape and size of particles with diameters ranging from 10 um to 6 mm. They make a distinction between a penetration that appears to be primarily composed of ash (pyrometeors) and penetrations composed of a mixture of pyrometeors and hydrometeors. This manuscript does an exceptional job of relating their unique and novel results to past literature and placing it in the context of auxiliary measurements. The manuscript is also exceptionally well-written, concise, and the figures are of very high quality. I recommend this manuscript be accepted for publication upon addressing a few minor comments listed below.

**Minor Comments:**

- Fig. 1b: What is the dBZe threshold used to define echo top?
-
  Echo tops in this figure use a threshold of 18 dBZe, which is the threshold employed for the NWS echo-top product. Labeling for this figure has been modified to indicate this threshold and a reference (Lakshmanan et al. 2013, Weather and Forecasting) has been added to the text.

  Fig. 1: panel labels (a-d) should be bigger

  The minimum font size on almost all figures has been increased.

- Lines 292-294: I do think it is interesting that the distribution shapes of N^ and A^ are different, but can you really say anything about comparing the order-of-magnitude

ranges when the units are not the same? I'm not sure that yields any relevant information.

The order-of-magnitude comparison between N^ and A^ has been removed from the narrative. The three sentences in question have been reduced to two sentences as follows: "Most notably the size distribution of $\hat{A}$ is bimodal in nature. A secondary peak is evident in the 2D-SV and CIP data over the 150-400 µm diameter range."

- I understand to some degree the interpretation of TWC vs. LWC in terms of collection efficiency from the Nevzorov probes. However, lines 473-475 are a bit confusing. I don't understand why an overestimate of cloud droplet effective radius by the PVM-100 probe leads to a smaller collection efficiency. Is this a reasoning that's due to the different techniques of PVM-100 vs. the Nevzorov probes or are you using the PVM-100 as a supplement to your conclusion? This reasoning in general could stand to be restructured as it is also hard to understand the paragraph that starts on line 480 regarding IWC without a little more context.

The last two paragraphs of section 3.2 have been significantly revised to increase clarity. Below is the revised text:

"While particles less than 50 µm in diameter during penetrations 2-6 are likely cloud droplets, the nature of larger particles is unclear since ice particles, pyrometeors and a combination of the two are plausible explanations. Pyrometeors in the form of ash particles could be lofted upward to the tops of the pyroCus, especially in penetrations 2-3 where updrafts larger than 1 m s$^{-1}$ are observed (Fig. 14a-b). Likewise, ice particles are possible since air temperatures at flight level are well below freezing (Fig. 14f-j). Also, conditions of ice supersaturation are evident in four of the five penetrations (Fig. 14f-i). Finally, the distributions of $A_r$ and $F_r$ in Figs. 10b-f and 11b-f, respectively, are consistent with values expected for ice particles. If these larger particles are composed of ice, they should produce a signal in the Nevzorov-TWC ($NEV_{TWC}$) data whereas ash particles should not be detectable by the Nevzorov probe. However, this analysis is complicated by the presence of small cloud droplets. Cloud droplets also produce a signal in the $NEV_{TWC}$ data, but the magnitude of that signal depends not only on the water content, but also on the size of the droplets themselves. For the same liquid water contents, the signal in $NEV_{TWC}$ will increase by ~50% as droplet effective radii increase from ~5 to ~15 µm, while over the same range of radii, the signal in $NEV_{LWC}$ only increases by ~2% (Fig. 4 and Equation 13 of Korolev et al., 1998). Thus, clouds composed of liquid, even when devoid of ice, will have different signals in $NEV_{TWC}$ and $NEV_{LWC}$ depending on droplet size regardless of liquid water content.

Values of $NEV_{TWC}$, $NEV_{LWC}$ and the ratio of these signals ($NEV_r = NEV_{TWC}/NEV_{LWC}$) are presented for penetrations 2 through 6 in Fig. 16. Following the discussion in the previous paragraph and the definition of $NEV_r$, an increase in $NEV_r$ may result from either (or both) an increase in droplet effective radius and/or an increase in IWC. In penetration 3, $NEV_r$ is 0.36 (Fig. 16b). Assuming the cloud is devoid of ice, this value corresponds to a droplet effective radius of ~2.5 µm. If, on the other hand, ice was present, then the radius would be even smaller. This radius is about half of that measured by the PVM-100 probe across all penetrations (Fig. 14 k-o) suggesting that the PVM may be overestimating the droplet effective radius by a factor of 2. $NEV_r$ varies from 0.36 to 0.69 across all five penetrations. If cloud droplet effective radii are assumed to be the same across these penetrations, then the variations in $NEV_r$ might be explained by the presence of ice particles. To further investigate the possible presence of ice, an independent measure of IWC is needed. Size distributions of OAP number concentration at 1 Hz are integrated over diameter with a mass-diameter relationship for ice-crystal aggregates from Brown and Francis (1995) to derive IWC ($OAP_{IWC}$). The derivation is restricted to particles having diameters greater than or equal to 50 µm to eliminate the contribution of cloud droplets.

Results of this analysis show that the smallest $OAP_{IWC}$ occurs during penetration 3, which coincides with the smallest $NEV_r$ (Fig. 16b). The largest and next largest $OAP_{IWC}$ are observed in penetrations 4 and 6, respectively where $NEV_r$ is largest (Fig. 16c,e). These trends suggest that ice particles larger than 50 μm diameter are present in the data produced by the Nevzorov probe and OAP's. However, this analysis does not eliminate the possibility that pyrometeors are also in the sample volume. Indeed, many of the measurements from penetration 3 show more similarity to those for the smoke/ash plume in penetration 1 than those for the pyroCu in penetrations 2 and 4-6. This suggests that the particles sampled during penetration 3 might be composed of a greater proportion of pyrometeors than the particles sampled during the other pyroCu penetrations."

**Line-specific comments:**

- Line 128: Maybe mention what the native sampling resolution is
-

  This sentence has been modified as follows: "This study uses versions of these parameters temporally degraded to 1 Hz from raw data sampled at 100-1000 Hz."

  Lines 165-167: Where did this 18 dBZe threshold come from?

  The 18 dBZe threshold is specified by the NWS in its radar echo top product. A relevant reference has been added for context (Lakshmanan et al. 2013, Weather and Forecasting)

- Line 212: "…and a large as…" should be "…and as large as…"

  Corrected

- Lines 279-280: This sentence is worded awkwardly because there is no overlap between the 2D-P and the other instruments. I would reformat the sentence to immediately imply a lack of overlap.

  This sentence has been modified as follows: "In contrast, there is essentially no overlap of mean $\widehat{N}$ from the 2D-P with mean $\widehat{N}$ from the 2D-SV and CIP."